# Tertiary lymphoid structures critical for prognosis in endometrial cancer patients

Nanda Horeweg [1,16✉], Hagma H. Workel[2,16], Dominik Loiero [3], David N. Church[4,5], Lisa Vermij[6],
Alicia Léon-Castillo[6], Ricki T. Krog [6,7], Stephanie M. de Boer[1], Remi A. Nout[8], Melanie E. Powell[9],
Linda R. Mileshkin[10], Helen MacKay[11], Alexandra Leary[12], Naveena Singh[13], Ina M. Jürgenliemk-Schulz[14],
Vincent T. H. B. M. Smit[6], Carien L. Creutzberg [1], Viktor H. Koelzer [3,15], Hans W. Nijman[2], Tjalling Bosse[6],
Marco de Bruyn [2] & TransPORTEC consortium*

B-cells play a key role in cancer suppression, particularly when aggregated in tertiary lymphoid structures (TLS). Here, we investigate the role of B-cells and TLS in endometrial cancer (EC). Single cell RNA-sequencing of B-cells shows presence of naïve B-cells, cycling/germinal center B-cells and antibody-secreting cells. Differential gene expression analysis shows association of TLS with L1CAM overexpression. Immunohistochemistry and co-immunofluorescence show L1CAM expression in mature TLS, independent of L1CAM expression in the tumor. Using L1CAM as a marker, 378 of the 411 molecularly classified ECs from the PORTEC-3 biobank are evaluated, TLS are found in 19%. L1CAM expressing TLS are most common in mismatch-repair deficient (29/127, 23%) and polymerase-epsilon mutant EC (24/47, 51%). Multivariable Cox regression analysis shows strong favorable prognostic impact of TLS, independent of clinicopathological and molecular factors. Our data suggests a pivotal role of TLS in outcome of EC patients, and establishes L1CAM as a simple biomarker.

[1] Department of Radiation Oncology, Leiden University Medical Center, Leiden, the Netherlands. [2] Department of Gynaecologic Oncology, University Medical Center Groningen, Groningen, the Netherlands. [3] Department of Pathology and Molecular Pathology, University Hospital Zurich, University of Zurich, Zurich, Switzerland. [4] Wellcome Centre for Human Genetics, University of Oxford, Oxford, United Kingdom. [5] Oxford NIHR Comprehensive Biomedical Research Centre, Oxford University Hospitals NHS Foundation Trust, Oxford, United Kingdom. [6] Department of Pathology, Leiden University Medical Center, Leiden, the Netherlands. [7] Department Surgery, Leiden University Medical Center, Leiden, the Netherlands. [8] Department of Radiotherapy, Erasmus MC Cancer Institute, Rotterdam, the Netherlands. [9] Department of Clinical Oncology, Barts Health NHS Trust, London, United Kingdom. [10] Department of Medical Oncology, Peter MacCallum Cancer Centre, Melbourne, VIC, Australia. [11] Division of Medical Oncology and Hematology, Sunnybrook Odette Cancer Centre, Toronto, ON, Canada. [12] Department of Medical Oncology, Gustave Roussy, Villejuif, France. [13] Department of Pathology, Barts Health NHS Trust, London, United Kingdom. [14] Department of Radiation Oncology, University Medical Center Utrecht, Utrecht, the Netherlands. [15] Department of Oncology and Nuffield Department of Medicine, University of Oxford, Oxford, United Kingdom. [16] These authors contributed equally: Nanda Horeweg, Hagma H. Workel. *A list of authors and their affiliations appears at the end of the paper. ✉email: n.horeweg@lumc.nl

The molecular classification of endometrial cancer (EC) distinguishes four subtypes with validated prognostic impact: i) ultra-mutated EC with DNA-polymerase epsilon exonuclease domain mutations (POLEmut) with an excellent prognosis; ii) hypermutated EC with mismatch-repair deficiency (MMRd) with an intermediate prognosis; iii) copy-number-high EC with frequent TP53 mutations (p53abn) with an unfavorable prognosis; and iv) copy-number-low EC without a specific molecular profile (NSMP) with an intermediate prognosis[1,2]. We recently demonstrated that assessment of CD8$^+$ tumor infiltrating T-lymphocytes (TILs) improves prognostication beyond clinicopathological risk factors and molecular class[3]. In another study, we found that T-cell responses led to B-cell driven immune responses via the secretion of CXCL13, a key driver of B-cell recruitment[4]. Expression of CXCL13 was associated with the formation of B-cell aggregates in and around ECs in the presence of high endothelial venules (HEV), germinal B-cells centers and dendritic cells surrounded by a rim of T-cells[4]. This specific type of ectopic lymphoid formations are known as tertiary lymphoid structures (TLS)[5,6]. At TLS, local and systemic B- and T-cell responses against cancer are initiated and maintained[6,7]. TLS presence is associated with a reduced risk of recurrence and improved response to immune checkpoint blockade (ICB) in several cancers[7–13]. A recent study assessed presence of TLS in EC by a 12-cytokine signature and identified most TLS in POLEmut and MMRd EC[8]. TLS were associated with a significantly better prognosis, but independence of this effect of clinicopathological features and molecular class was not assessed[8]. Moreover, TLS assessment in this study relied on methodology that cannot easily be implemented in clinical trials and routine diagnostics. We aimed to find a simple biomarker for TLS and better understand the role and prognostic relevance of B-cells and TLS in the immunity against EC. Here, we show intratumoral presence of germinal center/Cycling B cells and plasmablasts suggesting similarities with viral- and neoantigen-driven immune responses. Differential gene expression analysis shows association of TLS with L1CAM overexpression. Using L1CAM immunohistochemistry as a marker, TLS are found in 19% of high risk endometrial cancer patients and have a strong favorable prognostic impact of TLS, independent of clinicopathological and molecular factors.

## Results

**Role of B-cells in endometrial cancer**. We first performed an in-depth analysis of B-cell responses in EC by single-cell mRNA sequencing (scRNA-seq) of 1501 B-cells obtained from 6 ECs (Fig. 1A). Unsupervised clustering identified three main clusters of B-cells characterized by: naïve B cell genes including SELL and TCL1A (cluster 1); (pre-)Germinal Centre (GC)-like genes BHLHE40, BCL6, MKI67 and HMGB2 (cluster 2); and plasma cell genes PRDM1, XBP1, MZB1, and SSR4 (Cluster 3) (Fig. 1B; Supplementary Data 1). Individual patients were similarly represented within each cluster (Supplementary Fig. 1). To confirm the identity of cells in each cluster, we mapped the EC B cell dataset onto a reference dataset of tonsillar B cells (Fig. 1C, D)[14]. Cluster 1 cells resembled naïve B cells, cluster 2 a mixture of naïve, activated, cycling, memory B cells and plasmablasts, and cluster 3 naïve B cells and plasmablasts. We observed heterogeneous expression of immunoglobulin heavy and light chain genes (Supplementary Fig. 2) and examined these responses in more detail (Fig. 1E; Supplementary Fig. 3). Plasmablast clusters were consistent with B-cell class switching, and largely based on differential expression of heavy chain IGHG, IGHA genes and the mutually exclusive light chain IGLC and IGKC genes (Fig. 1E). Differential gene expression analysis revealed that these IgG and

IgA plasmablasts were transcriptomically similar (Supplementary Data 2). The intratumoral presence of GC/Cycling B cells and plasmablasts are in line with a recent study in HPV-associated HNSCC[15], and suggest similarities between B-cell responses in viral- and neoantigen-driven immune responses.

**Discovery of L1CAM expression in mature TLS**. The presence of ABC and GC B-cells raised the possibility that these could be a result of ongoing TLS formation. To confirm this, we quantified TLS using H&E-stained histological sections from The Cancer Genome Atlas (TCGA) uterine corpus endometrial cancer cohort (Fig. 2A). In line with previous observations[8,16], TLS were more common among the neoantigen-rich MMRd (20.6%) and POLEmut (30.3%) EC-subtypes (p53abn 2.9% and NSMP 2.9%, $p < 0.0001$) (Fig. 2B). Differential gene expression analysis revealed a significantly higher expression of genes associated with CD8$^+$-T-cell infiltration (CD8A) and effector function (LAG3, CCL5, NKG7, GZMH) in ECs with TLS (Supplementary Data 3). Gene set enrichment analysis identified that gene networks associated with T and B cell immunity were upregulated in TLS-positive vs. TLS-negative tumors (Supplementary Fig. 4 and Supplementary Data 4). Surprisingly, we also noted significantly greater expression of L1CAM in ECs with TLS (Fig. 2C and Supplementary Data 3). L1CAM is associated with increased metastatic potential of cancer cells but was also recently reported as a marker for follicular dendritic cells (FDCs)[17,18]. Accordingly, we used immunohistochemistry (IHC) and observed strong L1CAM staining in the germinal center of mature TLS (Fig. 2D), which co-localized with follicular dendritic cell (FDC) marker CD21 (Fig. 2E). This expression of L1CAM on FDCs was independent of L1CAM overexpression by the tumor. Analysis of sequential tissue sections confirmed presence and typical distribution of TLS hallmark immune cell subsets in TLS containing L1CAM-positive FDCs (Fig. 3).

**Concordance study of L1CAM as a marker for mature TLS**. To explore whether L1CAM IHC could be used as a marker for mature TLS, two pathologists (T.B. and V.H.K.) quantified mature TLS on H&E and L1CAM stained whole tumor slides of 50 ECs of the TransPORTEC biobank (Supplementary Tables 1 and 2). Uncertainty in the distinction between a lymphoid aggregate and a TLS was reported in 26% using H&E-stained slides, while L1CAM expression was never found in lymphoid aggregates that did not have the morphology of a mature TLS. The number of mature TLS per slide detected by H&E was systematically lower than by L1CAM (mean 1.1 vs. 1.8) and concordance between these two methods was moderate (intraclass coefficient 0.79, 95%CI 0.63–0.89; kappa 0.64, SE 0.11). The interobserver agreement using L1CAM to detect mature TLS was excellent (intraclass correlation coefficient 0.94, 95%CI 0.88–0.97; kappa 0.84, SE 0.8).

**Clinicopathological correlations of mature TLS**. To assess the clinicopathological correlations of L1CAM expressing mature TLS, we retrieved all L1CAM-stained whole tissue slides from the PORTEC-3 trial biobank[19]. In 378 of the 411 molecularly classified ECs L1CAM-stained slides of sufficient quality were available for TLS assessment (Supplementary Fig. 5, Supplementary Table 3). Using this method, TLS were observed at the tumor invasive border and in the myometrium in 71 of the 378 EC tissues (18.8%), ranging from 1 to 20 TLS per slide (median 2). As in the TCGA cohort, presence of TLS was associated with the neoantigen-rich POLEmut and MMRd molecular subtypes (Fig. 3A, Supplementary Table 4). Moreover, in ECs with multiple classifying features, such as MMRd EC with secondary p53-abnormality, TLS were more common (Fisher Exact $p = 0.009$) and more abundant

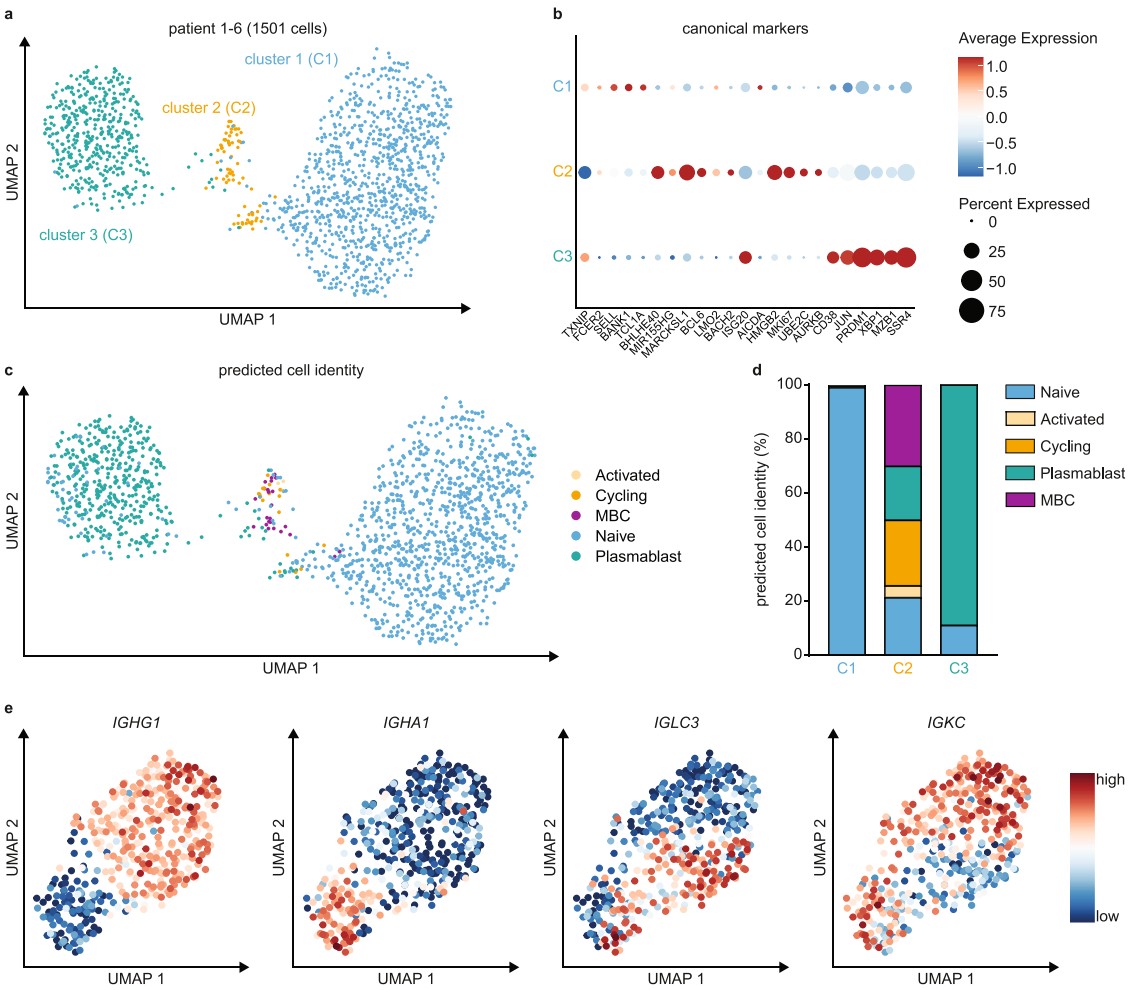

**Fig. 1 Single cell RNA sequencing of tumor-infiltrating B-cells. a** UMAP projection of endometrial cancer B-cell scRNA-seq data (1501 cells; 6 donors) annotated by cluster. **b** Dotplot of canonical B cell subtype marker genes per cluster. **c** Predicted cell identity for Cluster 1, Cluster 2 and Cluster 3 cells assigned using reference scRNA-seq data of a human lymph node. Cell identities are projected onto the endometrial cancer B-cell UMAP from **a**. **d** Quantification of predicted cell identities per cluster. **e** UMAP projection of Cluster 3 plasmablasts with Feature Plots depicting IGHG1, IGHA1, IGLC3 and IGKC expression in single cells. UMAP Uniform Manifold Approximation and Projection; MBC Memory B cell. Source data are provided as a Source Data file.

(Mann-Whitney $p = 0.013$, Table S4). While $CD8^+$ and $CD20^+$ densities were also significantly higher among POLEmut and MMRd ECs, only a subset had TLS (Fig. 3A, Supplementary Tables 5–8). Both the intraepithelial and the intrastromal densities of $CD8^+$ T-cells showed a significant and independent correlation with TLS presence, while only the intrastromal density of $CD20^+$ B-cells was weakly associated with TLS presence and lost significance after correction for clinicopathological features and molecular class (Supplementary Table 9).

**Prognostic impact of tumor-infiltrating T and B cells and mature TLS.** Next, we assessed the prognostic impact of $CD8^+$ and $CD20^+$-cell densities and TLS among PORTEC-3 participants. For $CD8^+$, both the intraepithelial and the intrastromal cell densities were strongly associated with a lower risk of recurrence (per doubling of density: HR 0.85, 95%CI 0.78–0.93, $p = 0.00049$ and HR 0.87, 95%CI 0.80–0.95, $p = 0.0046$ respectively), while only intrastromal $CD20^+$-cell density was significantly associated with a lower recurrence risk (HR 0.92, 95%CI 0.85–1.00, $p = 0.048$; Supplementary Table 10). Both the number and presence (none vs. ≥1) of TLS were strongly associated with a reduced risk of recurrence (HR 0.62, 95%CI 0.42–0.92, $p = 0.017$ and HR 0.25, 95%CI 0.10–0.62, $p = 0.0028$). Based on the effect size and model fit

(Supplementary Table 10) and the significant correlations between variables (Supplementary Table 11) we decided to proceed with the dichotomous TLS variable.

**Independent prognostic value of mature TLS.** Presence of TLS was a significant favorable prognostic factor for both time to endometrial cancer recurrence (Fig. 4B) and endometrial cancer-related death (Fig. 4C). Five-year risk of recurrence was 7.2% (95%CI 0.9–13.1%) for EC patients with TLS compared to 32.6% (95%CI 27.1–37.7%) for those without TLS. To determine whether prognostic impact of TLS was independent, we built a multivariable Cox proportional hazards model including the molecular classifier and all relevant clinicopathological features of high-risk EC (according to Léon-Castillo et al.[2], Table 1). TLS was a significant and independent favorable predictor of recurrence (HR 0.32, 95%CI 0.14–0.73, $p = 0.0073$) and endometrial cancer-specific survival (HR 0.15, 95%CI 0.04–0.61, $p = 0.0085$; Table 1, Supplementary Table 12). Addition of TLS to the prognostic molecular model as published by Léon-Castillo et al. significantly improved model fit and increased the total explained variance (AIC 1202.2 vs. 1194.2, C index 0.714 vs. 0.729 and likelihood ratio test for nested models $p = 0.0016$; Fig. 5A). Interestingly, the addition of TLS to the model did not change the relative

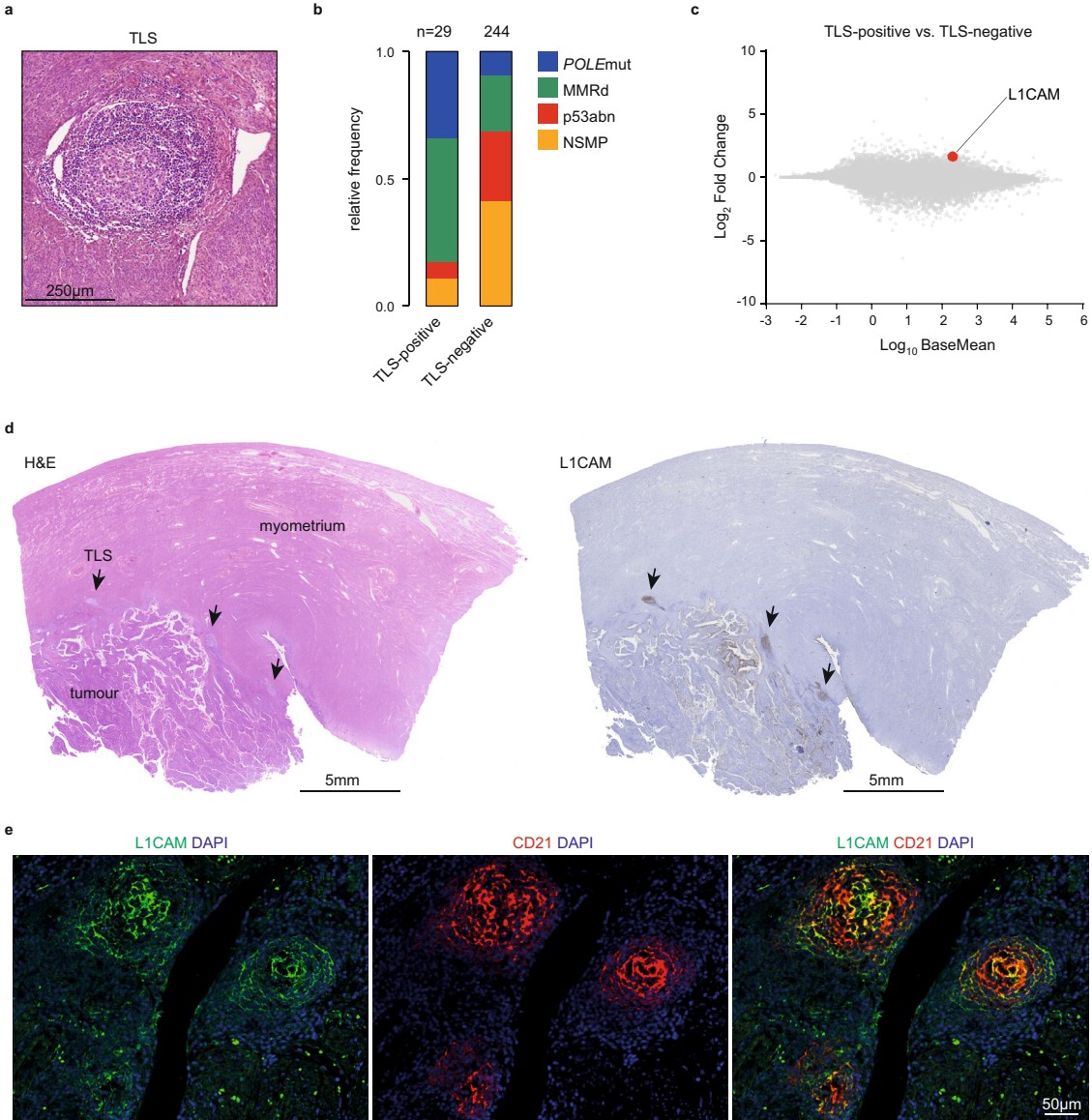

**Fig. 2 L1CAM expression in mature germinal centers of tertiary lymphoid structures. a** Representative H&E image of a TLS in EC, which were observed in 29 of the 273 cases of the Uterine Corpus Endometrial Carcinoma (UCEC) cohort of The Cancer Genome Atlas (TCGA) research consortium. **b** Frequency of molecular subgroups by TLS status in endometrial cancer patients included in the UCEC TCGA. Source data are available at: https://portal.gdc.cancer.gov. **c** Differential gene expression of TLS-positive versus TLS-negative TCGA UCEC cases. **d** Representative L1CAM-positive TLS case. Arrows indicate TLS. **e** Co-immunofluorescent analysis of L1CAM-positive TLS with L1CAM and CD21. The experiment was repeated four times with similar results. n Number of cases, *POLE*mut Pathogenic polymerase epsilon mutation, MMRd Mismatch repair-deficient, p53abn p53 abnormal, NSMP No specific molecular profile, TLS Tertiary lymphoid structure, L1CAM Ligant-1 cell adhesion molecule, DAPI 4′,6-diamidino-2-phenylindole. Source data are provided as a Source Data File.

contribution of the clinicopathological predictors much, but it reduced the contribution of the molecular classifier (Fig. 5B). Explorative subgroup analysis by molecular group (Fig. 6A, B) showed a significant favorable prognostic impact of TLS in MMRd EC ($p = 0.003$).

**Confirmation of TLS as the immunological marker of choice**. To verify whether TLS, rather than CD8$^+$ or CD20$^+$ densities, are the best addition to clinicopathological factors and molecular class in the prediction model, we performed a sensitivity analysis. We compared TLS with the strongest immune cell density markers for CD8$^+$ (intraepithelial) and CD20$^+$ (intrastromal). Using cases with all 3 immuno-biomarkers available ($n = 252$), 3 multivariable Cox proportional hazards models were built in

resemblance of the final molecular-immune model (Fig. 4A). These models showed (Supplementary Table 13) that intrastromal CD20$^+$ density had no independent prognostic impact after correction for clinicopathological factors and molecular class (HR 0.98, 95%CI 0.91–1.07, $p = 0.69$). Intraepithelial CD8$^+$ density had independent prognostic impact (HR 0.89, 95%CI 0.80–0.99, $p = 0.029$) but model fit was less good (concordance-index 0.736, se 0.027) than with TLS (0.745, se 0.026); likelihood ratio-test for nested models $p = 2.20 \times 10^{-16}$.

**Discussion**

In this study, we leveraged scRNA-seq of B-cells in EC to establish the presence of cycling/germinal center B-cells and antibody-secreting B-cells. The antibody-secreting B-cells had

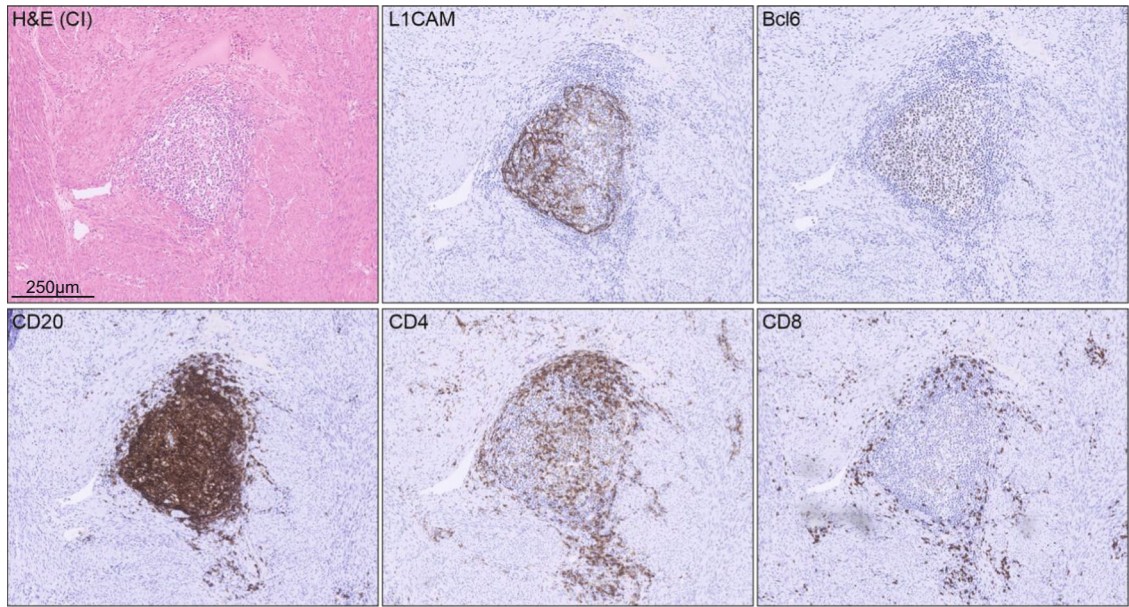

**Fig. 3 Hallmarks of mature TLS in L1CAM positive TLS.** Representative example of a single endometrial cancer case showing hallmark features of TLS maturation in L1CAM-positive TLS as determined by Bcl6, CD20, CD4 and CD8 immunohistochemistry.

undergone class-switching, suggesting TLS formation and an ongoing B-cell response against EC. Differential gene expression analysis showed an association of TLS with CD8[+] T-cell infiltration and L1CAM overexpression. IHC analysis of L1CAM-stained whole tumor slides showed ectopic lymphoid structures at the tumor invasive border and the myometrium that expressed L1CAM independent of any L1CAM expression by the tumor itself. The L1CAM-expressing lymphoid structures appeared to be mature TLS with a germinal center, based on co-immunofluorescence and IHC for hallmark immune cell subsets. Using L1CAM expression at lymphoid structures as a marker, we assessed tumor material of 378 high-risk EC patients included in the PORTEC-3 trial and found TLS in 19% of cases[19]. Subsequent analyses confirmed the favorable prognostic impact of TLS in an independent randomized trial with high quality clinical outcome data (PORTEC-3), which was previously only demonstrated in the TCGA dataset[8] and a small retrospective study[20]. We now also demonstrate that presence of TLS remains a strong favorable prognostic factor after correction for all important clinicopathological and molecular risk factors. Moreover, our analyses suggest that presence of mature TLS is an important factor determining that *POLE*mut and part of the MMRd EC patients have a favorable prognosis. Recent evidence shows that the maturity of a TLS and the presence of a germinal center in particular, is pivotal for its' prognostic impact[21]. In our study, L1CAM expression was only found in morphologically mature TLS with a germinal center. Potentially, L1CAM is a marker specific for the clinically relevant TLS.

The prognostic impact of TLS has only partially been explained in endometrial and other cancers. It is known that TLS orchestrate a specific and coordinated immune reaction that results in a high density of mature dendritic cells, tumor-infiltrating lymphocytes (TILs) and effector-memory CD8[+] T-cells[7,21]. T-cell:B-cell interactions in the TLS contribute to T-cell activation and maturation of B-cells to antibody producing plasma cells[5,7,22]. In addition, immunologic memory is generated that can mediate systemic immune surveillance against metastasis[21].

Knowledge of the specific conditions that promote TLS formation is important to advance towards identification of targetable mechanisms. Specifically, the observation that TLS are more

frequently present in *POLE*mut and MMRd ECs by us and others[8] supports the hypothesis that TLS may form in reaction to immunogenic tumor neoantigens, which are more likely to be present in cancers with high mutational burden[8]. The observation that TLS were relatively common among ECs with multiple classifying features may support this hypothesis because recent work in transgenic *POLE*mut mice suggested that co-occurring MMRd or *TP53* mutations help *POLE*mut cancer cells to cope with a high mutational burden and may drive a higher neoantigen load[23,24]. However, the fact that we also observed TLS in p53abn and NSMP EC suggests that conditions favorable for TLS formation can also occur, though infrequently, in cancers with a relatively low mutational burden.

The observed co-localization of L1CAM and CD21 may also shed light on the formation of TLS. CD21 demarcates FDCs, which are thought to originate from perivascular precursor cells, that undergo activation and maturation in response to lymphotoxin (LT) beta receptor signaling. As perivascular cells use L1CAM to migrate across the endothelial basal lamina, it is tempting to speculate that L1CAM-positive perivascular cells may represent FDC precursors in human tumors. A sequence of events could be envisioned where tumor-reactive T-cells release CXCL13[4,23], attracting CXCR5 + LT + immune cells to the perivascular space initiating L1CAM-positive perivascular cell activation and maturation to FDCs. Subsequent production of CXCL13 and inflammatory cytokines, chemokines and upregulation of cell adhesion molecules[25–27] including ICAM-2/3, VCAM-1 and MAdCAM-1 would mediate recruitment and adhesion of additional lymphocytes via high endothelial venules[27]. Together with recruitment of other immune cells such as dendritic cells and innate lymphoid cells, stroma maturation and stabilization into lymphoid stroma is promoted and TLS may form[25,26].

Bringing insights into the tumor microenvironment back to clinical practice is challenging, as the analysis methods used in studies are often not available in routine diagnostics. For example, gene signatures identified from transcriptomic analysis, such as the 12-cytokine signature for TLS[8,22], have been validated for TLS identification but are resource-demanding and difficult to implement. At the same time, use of H&E-stained slides has been shown to be poorly reproducible between pathologists[28]. Combined

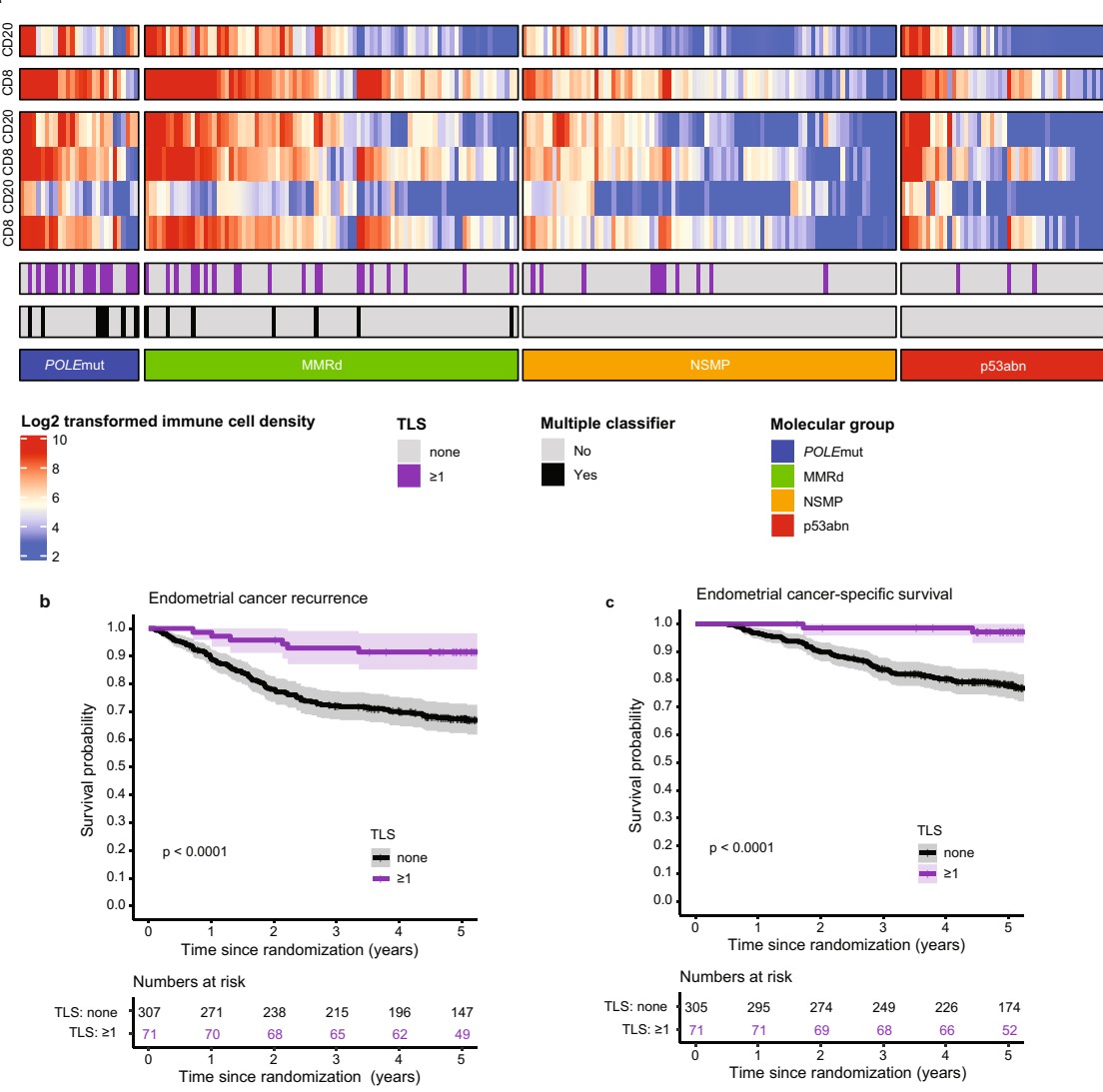

**Fig. 4 Relation between molecular group, TLS and CD8 densities and prognosis in high-risk endometrial cancer. a** Heat map of included PORTEC-3 patients ($N = 378$) with available data on molecular classification, TLS, CD8$^+$ and CD20$^+$ densities ($N = 252$). Each patient is represented by a row in the graph. Clustering of CD8$^+$ and CD20$^+$ densities stratified by molecular group was done by hierarchical clustering using Ward's minimum variance method. **b** Endometrial cancer recurrence-free survival calculated according to Kaplan-Meier's methodology using the log rank test (two-sided alpha of 0.05) for all included PORTEC-3 patients ($N = 378$). **c** Endometrial cancer-specific survival calculated according to Kaplan-Meier's methodology using the log-rank test (two-sided alpha of 0.05) for all included PORTEC-3 patients ($N = 378$).

immunohistochemical stains of TLS-hallmark immune cell subsets can be performed, but are inconvenient to quantify[11]. In contrast, TLS detection by a single immunohistochemical staining for L1CAM has high interobserver agreement and is simple to implement in clinical trials and routine diagnostics.

To conclude, we here demonstrate that presence of mature tertiary lymphoid structures, as assessed using L1CAM immunohistochemistry, improves prediction accuracy of recurrence and death beyond clinicopathological risk factors and molecular class in high-risk endometrial cancer patients.

## Methods

All included patients from the PORTEC-3 study have provided written informed consent for the use of their tumor material and data for this study. The study protocol was approved by the Ethics Committees of all participating groups, the details and study protocol are available online at: http://msbi.nl/portec3. For the single cell RNA sequence analysis, according to Dutch law, no approval from our institutional review board (Medisch Ethische Toetsingscommissie Leiden Den Haag Delft) was needed.

**Patient material**. The scRNA-seq analysis were performed using endometrial cancer digests obtained from surgical waste material of patients treated at University Medical Center Groningen, the Netherlands in accordance with local medical ethical guidelines and after written informed consent. All material was processed and stored anonymously. For initial collection of this archival series, tumors were collected when the diagnostic biopsy established the presence of an endometrial cancer. No other selection criteria were used, nor were the cases selected based on the presence of TLS.

Clinical data and tumor material from high-risk endometrial cancer patients participating in the randomized PORTEC-3 trial have been used (ISRCTN14387080, NCT00411138). The design and results of the PORTEC-3 trial have been published[19]. Briefly, 660 patients were included (2006–2013) with: FIGO-2009 stage IA endometrioid endometrial cancer grade 3 with lymph-vascular space invasion; or stage IB endometrioid endometrial cancer grade 3; or endometrioid endometrial cancer stage II, IIIA, IIIB (parametrial invasion) or IIIC; or stage I to III endometrial cancer with serous or clear cell histology. Written informed consent has been obtained from all patients.

**Immunohistochemistry**. L1CAM-stained whole tumor slides were available in the PORTEC-3 biobank. For a previous study[17] the formalin-fixed, paraffin-embedded tissue blocks were cut into 4 µm slides and mounted on Starfrost slides. Endogenous peroxidases were inactivated by 0.3% $H_2O_2$/methanol. Antigen retrieval was

**Table 1 Prognostic factors for recurrence in high-risk endometrial cancer.**

| Recurrence | Pathologic model | | | Molecular model | | | Molecular-immune model | | |
|---|---|---|---|---|---|---|---|---|---|
| N = 378, 111 events | HR | 95% CI | p-value | HR | 95% CI | p-value | HR | 95% CI | p-value |
| Age | 1.05 | 1.02–1.07 | 0.00032 | 1.03 | 1.00–1.05 | 0.045 | 1.03 | 1.00–1.05 | 0.034 |
| Adjuvant treatment | | | | | | | | | |
| RT | reference | | | reference | | | reference | | |
| CTRT | 0.76 | 0.52–1.10 | 0.15 | 0.70 | 0.48–1.03 | 0.069 | 0.72 | 0.50–1.063 | 0.093 |
| Histograde | | | | | | | | | |
| EEC grade 1-2 | reference | | | reference | | | reference | | |
| EEC grade 3 | 1.24 | 0.76–2.02 | 0.40 | 1.23 | 0.71–2.14 | 0.45 | 1.37 | 0.79–2.37 | 0.27 |
| non-EEC | 1.42 | 0.89–2.27 | 0.14 | 0.94 | 0.53–1.70 | 0.85 | 1.02 | 0.57–1.82 | 0.96 |
| Stage (I-II vs. III) | 1.93 | 1.28–2.91 | 0.0016 | 1.90 | 1.27–2.85 | 0.0020 | 1.98 | 1.31–2.97 | 0.0011 |
| LVSI | 1.25 | 0.82–1.96 | 0.30 | 1.29 | 0.83–2.00 | 0.25 | 1.24 | 0.80–1.94 | 0.33 |
| Molecular group | | | | | | | | | |
| NSMP | | | | reference | | | reference | | |
| POLEmut | | | | no events | | | no events | | |
| MMRd | | | | 0.89 | 0.53–1.48 | 0.64 | 0.99 | 0.59–1.64 | 0.96 |
| p53abn | | | | 2.63 | 1.46–4.73 | 0.0012 | 2.42 | 1.35–4.32 | 0.0029 |
| TLS | | | | | | | 0.32 | 0.14–0.73 | 0.0073 |

Three Cox proportional hazards models to showing impact of respectively clinicopathological factors, clinicopathological + molecular factors, and clinicopathological + molecular factors + presence of tertiary lymphoid structures on time to recurrence. Covariates were pre-specified according to Léon-Castillo et al. (J Clin Oncol, 2020)[2]. The addition of the molecular classifier to the pathologic model was associated with an improvement in model fit evidenced by: (i) reduction in Akaike's information criterion (AIC) 1244.968 vs. 1202.158, (ii) increase in model concordance (C index 0.655 vs. 0.714, and (iii) likelihood ratio test for comparison of nested models $p = 1.43 \times 10^{-10}$. Likewise, the addition of TLS presence improved model fit: (i) AIC 1202.158 vs. 1194.242, (ii) C index 0.714 vs. 0.729, (iii) likelihood ratio test for nested models $p = 0.0016$.
HR Hazard ratio; CI Confidence interval; RT Radiotherapy; CTRT Chemoradiation; EEC Endometrioid endometrial cancer; LVSI Lymphovascular space invasion; NSMP No specific molecular profile; POLEmut Pathogenic polymerase epsilon mutation; MMRd Mismatch repair deficient; p53abn p53 abnormal; TLS Tertiary lymphoid structure.

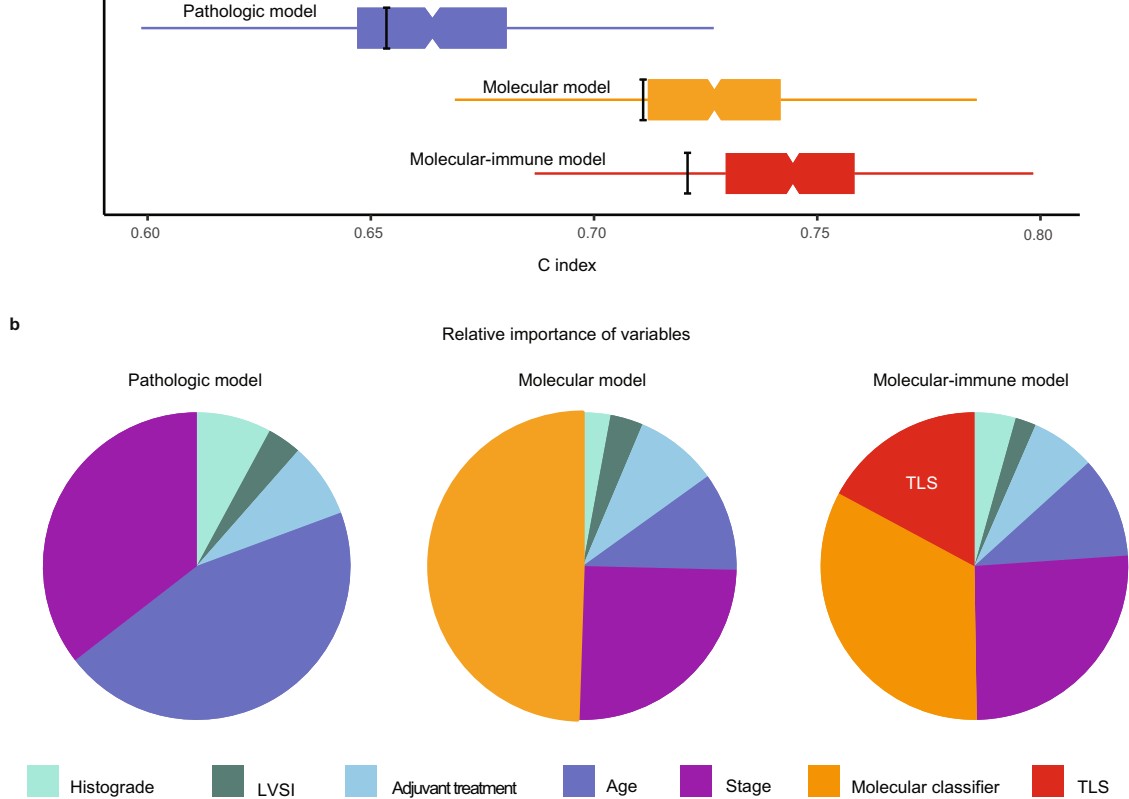

**Fig. 5 Characteristics of prognostic models in high-risk endometrial cancer. a** Boxplots showing concordance (C index) of the pathologic model, molecular model and the molecular-immune model. Box and whisker (Tukey) plots use results of 1000 bootstrap resamples from study population; lower and upper limits of box indicate 25th and 75th percentiles; and whiskers extend to 1.5x interquartile range below and above these values, respectively. The thick black bars indicate the C index from original population. **b** Pie charts showing relative importance of variables within these three multivariable models based on the proportion of the $\chi^2$ statistic. LVSI Lymphovascular space invasion; TLS Tertiary lymphoid structure.

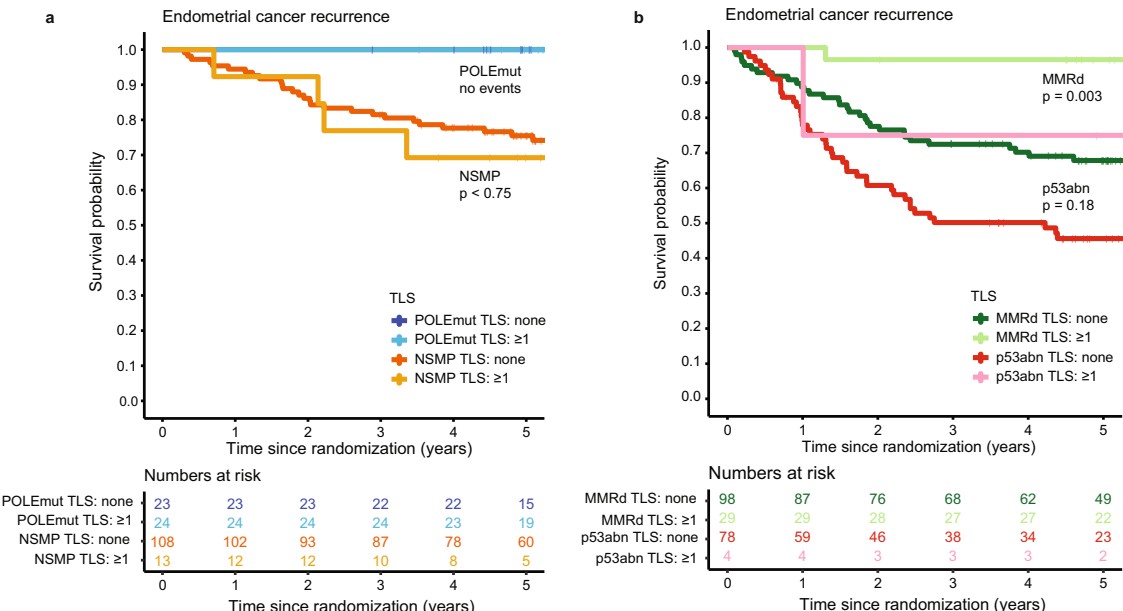

**Fig. 6 Endometrial cancer recurrence by TLS presence across the endometrial cancer molecular classes.** Endometrial cancer recurrence-free survival calculated according to Kaplan-Meier's methodology and tested between patients with and without TLS using a two-sided log rank test (alpha 0.05). **a** Patients with *POLE*mut endometrial cancer ($N = 47$) and MMRd endometrial cancer ($N = 127$). **b** Patients with p53abn endometrial cancer ($N = 83$) and NSMP endometrial cancer ($N = 121$). TLS Tertiary lymphoid structure; *POLE*mut Pathogenic polymerase epsilon mutation; MMRd Mismatch repair deficient; p53abn p53 abnormal; NSMP No specific molecular profile.

achieved by microwave oven treatment in 10 mmol/l Tris-EDTA, pH 9.0. Sections were incubated overnight with primary monoclonal antibodies against L1CAM (CD171; clone 14.10; 1:500; SIG-3911; Convance Inc.) For the assessment of $CD8^+$ and $CD20^+$ densities, TMAs were produced. This was done by marking morphologically representative areas of tumor on hematoxylin- and eosin-stained sections. Three core biopsies of 0.6 mm diameter each were randomly taken from the marked areas of tumor nests of the corresponding tissue block and placed in a recipient blank paraffin block (Shandon precision cut paraffin, No B1002490, Thermo) on pre-defined array locations, using a precision instrument (Beecher Instruments, Silver Spring, Maryland). When all cores were inserted, recipient blocks were placed in a 37 °C oven for 15 min in order to maximize adhesion of the cores to the surrounding wax. Four µm thick sections were cut from the TMAs. Immunohistochemistry for $CD8^+$ and $CD20^+$ was carried out on the TMAs. Antigen retrieval was performed in a preheated 10 mmol/L citrate buffer (pH = 6) and endogenous peroxidase activity was blocked by 0.45% hydrogen-peroxide. Slides were blocked in PBS containing 1% human serum and 1% BSA. Slides were incubated overnight with mouse anti-human CD8 (3 mg/L, clone C8/144B, GA62361–2, DAKO, Agilent Technologies) or anti-CD20 (0.63 mg/L; clone L26, catalog number M0755, Dako) at 4 °C. Subsequently, slides were incubated with a ready-to-use peroxidase-labelled polymer for 30 min (Envision+/HRP anti-mouse, K4001, Dako). Specific signal was visualized with 3,3′diaminobenzidin (DAB) and slides were counterstained with hematoxylin. Appropriate washing steps with PBS were performed in-between incubation steps. Sections were embedded in Eukitt mounting medium (Sigma Aldrich), and slides were scanned on a Hamamatsu digital slide scanner (Hamamatsu Photonics).

**Quantification of mature TLS using H&E and L1CAM IHC.** TLS were quantified by an expert gynecopathologist (T.B.) blinded for clinicopathological and molecular data. TLS assessment H&E-stained slides was based on morphology; rounded aggregates of organized lymphocytes in the myometrial wall or at the tumor invasive border were counted as mature TLS. For the assessment using L1CAM stained slides, TLS were counted if a rounded aggregate of lymphocytes with some level of organization was observed in the myometrial wall or at the tumor invasive border, with at least some (weak) L1CAM expression. Lymphoid aggregates without any L1CAM positivity were not counted.

**Concordance study of L1CAM for the detection of mature TLS.** A concordance study to explore whether L1CAM IHC could be used as a marker for mature TLS was conducted using L1CAM and H&E-stained whole tissue slides from the TransPORTEC biobank at Leiden University Medical Center. Fifty cases were randomly sampled with enrichment for the MMRd and *POLE* molecular classes. Firstly, an expert gynecopathologist (T.B.) counted the number of mature TLS on H&E and L1CAM stained whole tumor slides blinded for all clinicopathological and molecular characteristics of the cases. Secondly, another pathologist (V.H.K.)

also blinded, independently counted the number of mature TLS on L1CAM stained whole tumor slides. Intra- and interobserver agreement was expressed using the intraclass correlation coefficient for counts and Cohen's kappa value for the absence or presence of mature TLS.

**Machine learning-based $CD8^+$ and $CD20^+$ cell quantification.** Two pathologists (DL, VHK) performed digital slide review and quality control. Spots with staining artefacts, folds or <1000 cells/spot were excluded. Digital image analysis was performed using HALO digital image analysis software v3.0.311.355 (Indica Labs, Corrales, NM, USA). TMA slides were segmented into individual spots and linked to clinical information. By annotating tissue regions, a deep neural network algorithm was trained to localize and quantify tumor epithelial tissue and tumor-associated stroma regions. Classification accuracy was confirmed after generation of graphical overlays for each tissue area. Cell and staining quantification were carried out with following specifications: Nuclei (hematoxylin, RGB 20, 24, 65), CD8 (DAB, RGB 122, 93, 65), CD20 (DAB, RGB-102, 73, 60). Unstained epithelium and stromal fibroblasts served as internal negative controls. Marker positive cells were then allocated to the tumor or stroma compartment and infiltration density (cells/mm²), was recorded. Final scores for each case were calculated as the mean of the infiltration densities across all cores. Average densities were $\log_2$ transformed and negative values for $\log_2$ transformed densities were imputed with 0 before analysis to approximate a normal distribution. Concordance between pathologist estimation and artificial intelligence-based quantification has been demonstrated[3].

**Single-cell RNA sequencing.** Tumors were cut into ca. 1 cm³, enzymatically digested in RPMI medium (Gibco, Paisley, UK) with 1 mg/µL collagenase type IV (Gibco Life Technologies, Grand Island, USA) and 12, 6 µg/mL recombinant human DNase (Pulmozyme, Roche, Woerden, the Netherlands) for 30 min at 37 °C or overnight at room temperature. Digests were filtered using 70 µm cell strainers (Falcon) and enriched for peripheral blood mononuclear cells (PBMCs) using Ficoll-Paque PLUS (GE Healthcare Life Sciences, Marlborough, MA, USA). Cells were stored in liquid nitrogen until cell sorting. For cell sorting, tumor digests were thawed and washed with PBS and incubated with Zombie Aqua (1:100, Biolegend, San Diego, USA) for 15 min at room temperature. Samples were washed and stained with anti-CD27 APC-efluor 780 (clone O323; 47-0279-42; 0.025 µg/mL; eBioscience) and either CD19-PE (clone HIB19; 12-0199-41; 0.025 µg/mL; eBioscience), CD19-BV421 (clone HIB19; 562440; 50 µL/mL; BD Biosciences), CD19-APC (clone HIB19; 17-0199-41; 0.0125 µg/mL; eBioscience) or CD19-PE-cy7 (clone HIB19; 25-0199-41; 0.025 µg/mL; eBioscience) for 45 min at 4 °C. Cells were washed and filtered using a 35 µm strainer (Falcon). Patient samples were pooled to minimize differences due to plate effects and CD19-positive cells sorted on a Beckman Coulter MoFlo Astrios. Each well contained 2 µl lysis buffer (0.2% Triton X-100 (Sigma-Aldrich) and 2U RNase inhibitor (Takara)) with 1 µl 10 µM

oligo-dT primer and 1 μl 10 mM dNTP mix (Thermo Scientific). After sorting, the plate was spun down and incubated at 72 °C for 3 minutes. We used a modified SMARTseq2 protocol using custom-made primers. SmartScribe reverse transcriptase (Westburg-Clontech) and a template switching oligo (BC-TSO) were used to generate cDNA. Next, an exonuclease step was performed using 1:400 dilution of Exonuclease I. A PCR preamplification step was done with KAPA HiFi HotStart Ready Mix (Roche Diagnostics, 23 cycles in experiment 1 and 25 cycles in experiment 2) and a custom-made PCR primer. The cDNA samples were purified using Ampure XP beads (Beckman Coulter) in a ratio of 0.8:1 (Ampure bead:cDNA). Samples were analyzed on a 2100-Bioanalyzer using a PerkinElmer LabChip GX high-sensitivity DNA chip (Agilent) and on a Qubit™ 4 Fluorometer (ThermoFisher Scientific) according to manufacturer's instructions. 500 pg of each sample was tagmented and N7xx and S5xx index adapters were used for barcoding according to the Illumina Nextera XT DNA sample preparation kit (Illumina). Thereafter, samples were purified with Ampure XP beads (ratio 0.6:1 Ampure:cDNA) and analyzed on a 2100-Bioanalyzer. Samples were equimolar pooled (4 nM), and samples were sequenced on an Illumina Nextseq500 2500 using 75 bp paired end reads. The obtained mRNA sequencing data was demultiplexed into individual FASTQ files followed by alignment to the human reference genome hg38 using STAR (version 2.5.2)[29].

**Processing and annotation of single-cell RNA-seq**. Single-cell sequencing data was analyzed in R (Seurat (V4.0.4) package)[30]. Data were quality controlled, and size-factor normalized according to the single-cell workflow SingleCellExperiment using default settings (version 1.8.0)[31]. Data were transferred to Seurat, normalized (NormalizeData), scaled (ScaleData) and analyzed for annotation by running PCA (RunPCA), nearest neighbor graph (FindNeighbors) with 10 dimensions and unbiased clustering (FindClusters) with resolution set to 0.2. Uniform Manifold Approximation and Projection (UMAP) was used to visualize B cell clusters. Gene expression markers for different clusters of B cells were identified using the FindAllMarkers() command from Seurat with default settings, including Wilcoxon test and Bonferroni p-value correction. Plasmablasts were separated for a more detailed annotation by recomputing the PCA (RunPCA), nearest neighbor graph (FindNeighbors) and unbiased clustering (FindClusters). UMAP and FeaturePlots were used to visualize *IGHG*, *IGHA* and *IGLC* and *IGKC* clusters. A previously published tonsil scRNA-seq dataset (E-MTAB-9005)[14] was used to predict cell identity using the 'anchor'-based integration workflow from Seurat. Hereto, data was scaled and normalized using NormalizeData with default parameters, followed by the FindTransferAnchors() and MapQuery() commands to project EC B cells onto the tonsil scRNA-seq data.

**Differential gene expression analysis of UCEC TCGA data**. MRNA-seq and clinical data from uterine corpus endometrial carcinoma (UCEC) were downloaded from firebrowse.org. TLS were quantified using H&E images of the UCEC TCGA cohort downloaded from https://portal.gdc.cancer.gov. Differentially expressed genes were assessed between TLS-positive and TLS-negative MMRd and *POLE*-EDM cases by DESeq2 (version 2_1.30.0)[32]. Genes with a Benjamini–Hochberg FDR < 0.01 and log2 fold change > 1 were selected as significantly different. Gene Set Enrichment Analysis was performed using Cluster-Profiler (V4.0.5)[33,34] with GO terms for biological process. GO terms with a Benjamini–Hochberg FDR < 0.05 were selected as significantly different.

**Statistical analysis**. Biomarker analyses were performed in accordance with the REMARK guidelines and are listed in Supplementary Table 13. Primary endpoint was time to endometrial cancer recurrence, defined as time from randomization to recurrence, with censoring at last follow-up in case of no recurrence. Secondary endpoint was endometrial cancer-specific survival, defined as time from randomization to endometrial cancer death, with censoring at last follow-up in alive patients.

Continuous variables were analyzed using the T-test or Mann-Whitney U test depending on their distribution. Categorical variables were analyzed by the Mann-Whitey U test (ordinal variables) or the chi-square, Fisher's exact test or Fisher-Freeman-Halton test depending on the number of categories and expected events. Correlations were assessed using the Spearman correlation coefficient. Hierarchical clustering was used (Ward minimum variance method with Euclidean distances) to group CD8+ and CD20+ cell densities stratified by molecular class. Median follow-up was calculated by the reverse Kaplan–Meier method. Time-to-event analyses were performed by the Kaplan-Meier method, log-rank tests and multivariable Cox proportional hazards models. Clinicopathological and molecular variables for inclusion in multivariable models were prespecified based on the model published by Léon-Castillo et al.[2]. Model validation was performed by analysis of discrimination and indices of optimism determined by means of model fitting to 1000 bootstrap resamples. Proportionality of hazards was confirmed by inspection of scaled Schoenfeld residuals. Relative importance of variables within the multivariable models is based on the proportion of the $\chi^2$ statistic. Statistical significance was accepted at $p < 0.05$ (two-sided). Statistical analyses were performed in SPSS version 25 and R Version 3.6.3. using i.a. the following packages: ComplexHeatmap, rms, Survival, ggPlot2, Survminer, Hmisc and tidyverse.

**Reporting summary**. Further information on research design is available in the Nature Research Reporting Summary linked to this article.

## Data availability

The study protocol and other documentation of the PORTEC-3 trial are publicly available at http://msbi.nl/portec3. The tumour material and datasets generated during and/or analysed during the current study of the PORTEC-3 trial participants are not publicly available due to restrictions by privacy laws. Data and tumour material are currently available to the members of the international TransPORTEC consortium, and the consortium is open for requests for sharing of the data and material after receipt and evaluation of a scientific proposal. Requests should be addressed to the corresponding author within 15 years from the date of publication. Depending on the specific research proposal, the TransPORTEC consortium will determine when, for how long, for which specific purposes, and under which conditions the requested data can be made available, subject to ethical consent.

The TCGA-UCEC data used in this study are available in the National Cancer Institute database and publicly accessible via the GDC data portal [https://portal.gdc.cancer.gov/projects/TCGA-UCEC].

The raw scRNA-seq data generated in this study is available at NCBI Gene Expression Omnibus (GEO) under registration number GSE180091 and in the Source Data Files published alongside this article. The remaining data are available within the Article, Supplementary Information or Source Data file.

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

## Acknowledgements

We gratefully acknowledge the PORTEC-3 study group and TransPORTEC consortium for their contribution to the conduct of the PORTEC-3 trial and the establishment of the TransPORTEC biobank. A list of the members of these consortia is provided in the Supplementary Information. The authors would also like to thank Nienke van Rooij, Diana Spierings and Bart Eggen for their support in the acquisition of the single cell RNA sequencing data. This work was supported by the Dutch Cancer Society (project grants: LUMC KWF-2018-1-11629, RUG 2015-7235 and 8232-21648). This work was supported by The Oxford NIHR Comprehensive Biomedical Research Centre (BRC) and WHG core funding from the Wellcome Trust (203141/Z/16/Z). The views expressed are those of the authors and not necessarily those of the NHS, the NIHR, the Department of Health. This is an academic-sponsored, investigator-initiated study, the funding bodies had no role in the study design, data collection, analysis or manuscript writing.

## Author contributions

Design: N.H., D.C., V.K., H.N., T.B., M.d.B. Data and material: L.V., A.L.C., S.d.B., M.P., L.M., H.M., A.L., N.S., I.J.S., C.C., H.N., M.d.B. Experiments: H.W., D.L., D.C., R.K., V.K., T.B., M.d.B. Statistical analysis: N.H., D.L., D.C., V.K., M.d.B. Manuscript writing: N.H., M.d.B. Critical manuscript review: H.W., D.L., D.C., S.d.B., R.N., C.C., V.K., H.N., T.B.

## Competing interests

Dr. Horeweg reports outside of the submitted work to have received research grants from the Dutch Cancer Society (KWF-2021-13400, KWF-2021-13404). Dr. Church is funded by a Cancer Research UK Advanced Clinician Scientist Fellowship (C26642/A27963) and reports to be part of the advisory board for MSD. Prof. Nout, Dr. Bosse and Prof. Creutzberg report to have received a grant from the Dutch Cancer Society for the PORTEC-3 trial (KWF 2018-1-11629). Prof. Koelzer reports grants from Promedica Foundation (F-87701-41-01) during the conduct of the study and having served as an invited speaker on behalf of Indica Labs. Dr. de Bruyn reports, outside the submitted work, having received grants from the Dutch Cancer Society (KWF), grants from the European Research Council (ERC), grants from Health Holland, grants from DCPrime, non-financial support from BioNTech, non-financial support from Surflay, non-financial support from MSD, grants and non-financial support from Vicinivax. In addition, dr. de Bruyn has grants and non-financial support from Aduro Biotech, in part relating to a patent for Antibodies targeting CD103 (de Bruyn et al. No. 62/704,258). The remaining authors declare no competing interests.

## Additional information

## TransPORTEC consortium

Nanda Horeweg [1], David N. Church [4,5], Stephanie M. de Boer [1], Remi A. Nout [8], Melanie E. Powell [9], Linda R. Mileshkin [10], Helen MacKay [11], Alexandra Leary [12], Naveena Singh [13], Carien L. Creutzberg [1], Hans W. Nijman [2], Tjalling Bosse [6] & Marco de Bruyn [2]

