## [Peer Review File · Nature Communications]

Reviewers' Comments:

Reviewer #1:

Remarks to the Author:

The study reported in the manuscript by Horeweg et al., aimed to delineate the role and prognostic relevance of B-cells and tertiary lymphoid structures (TLSs) in endometrial cancer (EC). At the onset of this study, it was known that most TLSs in endometrial tumors occur in POLE-mutated (POLEmut) and mismatch repair deficient (MMRd) endometrial tumors, which are two of four major molecular subtypes of EC. The study by Horeweg et al., validates these prior observations and extends upon them to demonstrate the following:

- 1) "Antibody-secreting B-cells (in in endometrial tumors) had undergone class-switching and expressed markers of activation and exhaustion, suggesting TLS formation and an ongoing B-cell response against EC."
 - 2) "...an association of TLS with CD8+ T-cell infiltration and L1CAM overexpression..... independent of any L1CAM expression by the tumor itself."
 - 3) "L1CAM-expressing lymphoid structures appeared to be mature TLS with a germinal center, based on co-immunofluorescence and IHC for hallmark immune cell subsets. Using L1CAM expression at lymphoid structures as a marker, assessment of tumor material of 378 high-risk EC patients included in the PORTEC-3 trial revealed TLS in 19% of cases."
 - 4) "...favorable prognostic impact of TLS in an independent randomized trial with high quality clinical outcome data 171 (PORTEC-3)."
 - 5) The "presence of TLS remains a strong favorable prognostic factor after correction for all important clinicopathological and molecular risk factors."
- The authors conclude their "data suggests a pivotal role of TLS in outcome of EC patients, and establishes L1CAM as a simple biomarker."

While this is an interesting study, in its current form the manuscript is at times difficult to follow particularly regarding the exploration of L1CAM as a potential biomarker for the identification of TLS in EC.

Specific comments:

1. Lines 47-49 in abstract: Please include the percentages of MSI tumor and POLE-mutant tumors that were TLS positive; also, please state that these were L1CAM-positive-TLSs.
2. Lines 51-52: Last sentence of the abstract is overstated; the data in this study do not establish L1CAM as a biomarker of TLS. No data were presented regarding increased sensitivity and/or specificity of TLS identification using L1CAM-positivity compared with pathologic review of H&E sections. It would be more appropriate to state that this study highlights L1CAM is a potential biomarker for TLS in ECs.
3. Lines 85 & 100: More information is needed about the molecular and clinicopathologic criteria used to select the six EC tumors for scRNA-seq of B-cells, and the possible biases that the selection criteria might have introduced when defining the scRNA-seq clusters. In addition, were TLS(s) present in these six cases, and was this part of the selection criteria?
4. Lines 116-118: The authors refer to "L1CAM-stained whole tumor slides". Please define "L1CAM stained." Does this refer to slides that had L1CAM-positive staining of tumor cells, or does it refer to slides stained for L1CAM regardless of the level of L1CAM expression detected?
5. What percentage of L1CAM-negative tumors have TLS?
6. Line 103-104 states : "TLS were more common among the neoantigen-rich MMRd and POLEmut EC-subtypes (Figure 2B)." It would be important to provide the percentage of each molecular subgroup that is TLS+, along with p-values to indicate whether the differences in TLS incidence between molecular subgroups are statistically significant.
7. Lines 108-111 states: "We followed up on this observation by performing L1CAM-immunohistochemistry (IHC) on EC samples (Figure 2D). We observed strong L1CAM staining in the GC-like structures of the TLS, which co-localized with follicular dendritic cell (FDC) marker CD21 (Figure 2E). This was independent of L1CAM overexpression by the tumor." These statements of results are very generalized. More details are needed regarding (a) the number of EC samples assessed and the percentage of samples that were positive; (b) concordance rates of L1CAM positivity and GC-like structures of the TLS; and (c) the concordance rates of L1CAM+ in the TLS and L1CAM+ in the paired tumor sample.

8. Line 114 onwards: The rationale for exploring whether expression of L1CAM could be used as a marker for the presence of mature TLS is unclear. This part of the study seems circular in that the authors initially identified TLS in TCGA tumors by pathologic review of H&E stained tumor sections and subsequently identified differential (increased) expression of L1CAM levels in TCGA tumors with TLS. In other words, because the authors initially identified TLSs by pathologic review, it is unclear why a biomarker for TLS identification is needed. Assuming they can justify why a biomarker is warranted, the rationale for focusing on L1CAM as a potential biomarker rather than focusing on another significantly differentially expressed gene(s) highlighted in red in Figure 2C also requires justification.

Reviewer #2:

Remarks to the Author:

This is an interesting and well-presented study that investigates the roles of B cells and TLS in endometrial cancer. scRNA-seq was used to interrogate the various molecular subsets of TIL-B, revealing activated/memory B cells, GC B cells, and ASCs. TLS were associated with L1CAM expression, which in turn was investigated as a prognostic marker in a large (378 case) EC biobank (PORTEC-3). TLS were enriched in the POLE and MMRd molecular subtypes. L1CAM+ TLS were strongly favorably prognostic by multivariable analysis, independent of molecular subtype. The authors propose that TLS play a pivotal role in EC outcomes, and they propose L1CAM serves as a convenient biomarker for TLS.

Strengths:

1. B cells and TLS are the subject of much current interest.
2. scRNA-seq and spatial transcriptomics data enriches the emerging landscape of B cell phenotypes in human cancer.
3. EC provides a useful setting to evaluate B cells and TLS relative to distinct molecular subtypes.
4. Large and well annotated tissue cohort.

Limitations:

1. They provide a relatively superficial analysis of the scRNA-seq data, with no major discoveries.
2. The POLE and MMRd subtypes are known to be immunologically hot (including B cells and plasma cells; PMID: 30523022), therefore it is not surprising they are enriched for TLS.
3. TLS have been shown to associate with favorable prognosis in many cancers, therefore this finding is not novel, apart from providing a convincing example in EC.
4. The authors provide only a superficial validation of L1CAM as a surrogate biomarker for TLS. There is no comparison between the number of TLS one captures with canonical markers vs. L1CAM. And are they the same TLS with the same features? They show only one representative image. Moreover, their analysis is restricted to EC, leaving it an open question whether L1CAM is relevant to other cancers.

Suggestions for improvement:

1. For the scRNA-seq expts, which molecular subtypes of EC tumors were used? The authors claim that cluster 1, 2 and 3 presented in Figure 1A correspond to activated/memory B cells, cycling/GC B cells, and antibody-secreting cells respectively. With the differentially expressed genes highlighted in Fig. 1B, I am convinced that cells in cluster 3 are indeed antibody-secreting cells. However, the identity of the other two clusters remains unclear. The authors should provide a more in-depth analysis of canonical marker genes expected in each of these cell types. Perhaps using the DotPlot() function from Seurat to plot expression of canonical markers genes across clusters would be more convincing. The study by King et al, cited in the Methods section, does provide an extensive of marker genes that could be used here [PMID: 33579751].
2. Besides the marker gene analysis presented in Fig. 1B, the authors used spatial transcriptomics to help assign a cell type to each of the three clusters identified by scRNA-seq. However, the explanations provided in the manuscript (including the ones in the Methods section) are too vague, making it hard to understand what was actually done.

3. In Figure 1D, the authors argue that this cluster represents activated/memory B cells that might be exhausted given that they express both HLA-DRA and CD11C. However, very few cells seem to actually express these two genes, and the cluster seem rather heterogeneous based on the expression of the characteristic marker genes shown in Figure 1B. It might be useful to characterize that cluster further by re-analyzing it similarly to what was done for the antibody-secreting cells in the next panel.

4. If you redo the analysis presented in Figure 1E but masking the immunoglobulin genes, are there any other genes that emerge as differentially expressed between the clusters? Could they help identify plasma cell subtypes in EC? For instance, do the IgA+ plasma cells appear to be immunosuppressive as has been reported in murine studies [PMID 29144460][PMID 25924065]?

5. Ideally, all codes as well as processed scRNA-seq data should be made available for download using a GitHub repository or Zenodo. If possible, the raw data should also be made public in an appropriate repository or at least be made available upon request. Similarly, the authors should add Supplementary Tables with the differentially expressed genes identified in each comparison.

6. In Fig. 2A, exactly what criteria were used to quantify TLS in H&E stained sections? In Figure 2B, a Fisher's exact test should be used to assess statistical significance.

7. The observation made in Figure 2C is interesting. However, no comments are made on the genes upregulated in TLS-negative tumors. Are they linked to tumour progression? Moreover, rather than cherry picking the genes to highlight, it would be preferable to formalize the analysis using GSEA or the like. In terms of graphical representation, encoding the p-value with the size of the dot is difficult to read given the extent of overplotting. Consider using arrows to signal what is up and down-regulated, and use colors to highlight the most significant genes.

8. With regard to the data presented in Figure 2D and E, the authors should comment on the prevalence of these L1CAM-TLS since they mention in the Methods that 'Lymphoid aggregates without any L1CAM positivity were not counted'. Similarly, according to Supplementary Figure 2, L1CAM+ TLS show all the features of a typical TLS. Could this be quantified? That is, out of the L1CAM+ TLS stained with CD4, CD8, Bcl6, CD20 panel, how many have a typical architecture? Such quantifications are required before calling L1CAM a TLS biomarker.

9. In Fig. S3, exactly how were TLS assessed (i.e. what were the precise scoring criteria)? In Figure 3D and E, no p-value is reported.

10. Figure 4B/C are barely commented by the authors. What is an acceptable c-index? Can interesting conclusions be drawn by looking at the relative importance of the variable? Moreover, in Figure 4C, the authors should replace these pie charts by bar plots to make the comparison easier for the reader [PMID 24645191].

11. The authors should discuss why their findings differ from those of Talhouk [PMID: 30523022] with respect to the relative prognostic effects of TIL versus molecular subtype (including TIL-B).

Reviewer #3:

Remarks to the Author:

This study from Horeweg et al examines the composition, phenotypic markers and link with disease progression of tertiary lymphoid structures (TLS) in endometrial cancer (EC). The authors first examine the B cell compartment of ECs using single-cell RNA-seq (scRNA) and report distinct B cell phenotypes associated with activation, GC maturation or antibody production. They then leverage biobanked tissue resources to quantify the frequency of TLS structures in different classes of EC, and report a particular association between TLS and the POLEmut and MMRd ECs. Differential gene expression of TLS positive and TLS-negative ECs identified L1CAM as a histological marker of TLS, which they then used to build a prognostic model. From this, the authors propose that the existence of TLS is associated with favourable patient outcomes.

This is a high quality study that I found to be very well written and reasoned, and it tackles an important and interesting area of cancer immunology. As well as reporting a convincing association between TLS and patient outcome, the transcriptomic profiling of tumour-associated/TLS B cells is a valuable resource for the scientific community to help understand the formation, maintenance and function of TLS, which remain poorly understood. Furthermore, while TLS are known to play an important role in mediating immune responses to cancer (as convincingly presented and discussed in this manuscript), they are also a key feature in the pathology of varied autoimmune diseases (eg lupus, rheumatoid arthritis, Sjogrens syndrome). Thus, the dissection of the structure and function of TLS is relevant beyond EC and will be of interest to many immunologists.

My only major reservation relates to the precise phenotypic annotation of B cell states by scRNA-seq, which can be addressed with additional analysis and discussion.

Major comments/concerns

- Some of the markers used to describe or annotate B cell scRNA-seq clusters C1 and C2 should be re-examined. (The markers used to annotate C3 as antibody secreting cells/plasma cells are very convincing). The spatial mapping generally supports the other annotations, but some of the current markers used are not particularly convincing. I suggest that the authors examine the following:

- o Cell cycle stage (G1, G2M, S) – HMGB2 (a marker for C2 cluster) is most highly expressed by proliferating/cycling B cells. These may be cycling B cells (most likely GC B cells) that cluster separately very distinctly from other GC B cells that could be contained within C1

- o Line 87 - HMGB2, BHLHE40 and VIM are not classical markers for GC B cells. Please examine BCL6 as a canonical GC B cell marker, and other transcriptional markers like GMDS, BACH2, LMO2, SERPINA9, AICDA, ISG20.

- o Some markers for C1 such as EZR and FOXP1 are also expressed by GC B cells. It is often helpful to visualise key marker genes on the single cell UMAP plot (FeaturePlot in Seurat) and I recommend plotting some of the key canonical marker genes for different B cell states in this fashion.

- o Are there any memory B cells present as part of C1? Please examine TNFRSF13B (most expressed by memory cells) and TCL1A (expressed by naïve and GC B cells, but not memory or ASCs)

- o Other markers I would suggest to examine include for CD69 and JUN for activated B cells, FCER2 for basal naïve B cells.

- o Please provide a more complete list of marker genes per cluster as a supplementary table. Either the top 50 or 100 genes per cluster, or the direct output from Seurat's FindAllMarkers command.

- o I would recommend examining/visualising the antibody class expression (IgM, IgD, IgG, IgA, IgE) of all B cell clusters, not just ASCs as presented in Fig1E. This can be very useful to help annotate different B cell phenotypes and may reveal insights into whether memory B cells of different classes may be interesting or relevant to TLS/EC.

- Was the tonsil scRNA-seq data from King et al 2021 (that was used as a reference for spatial transcriptomics) also integrated with the TLS EC B cell scRNA dataset? If the B cell annotations from King et al were transferred/projected on to the data from this study (or vice versa) does this support the annotations presented in Figure 1? In my opinion, this would be more convincing evidence to support the annotations that the comparison with the spatial mapping that was performed (although this is also very valuable).

- Line 161-163 "The antibody secreting B-cells had undergone class-switching and expressed markers of activation and exhaustion, suggesting TLS formation and an ongoing B-cell response against EC." As presented, the ASCs (plasma cells) do not consistently express the activation markers or exhaustion (CD11c) this should be re-written. Do the authors mean the activated/memory B cells?

- The authors may like to refer to a recent analysis of the same human lymph node spatial transcriptomic dataset that they examined (Kleshchevnikov et al 2020 bioRxiv 10.1101/2020.11.15.378125). I believe that their conclusions broadly support the analysis presented here.

- Fig 1E - Did the authors examine differential expression between these clusters? I am not convinced that this clustering analysis represents any major biological insight. The same information (ie that ASCs contain both IgG and IgA populations) could be presented by visualising these genes on the same UMAP as in Fig1A (with the advantage of presenting information about the isotype of other B cell populations as well)

- Relating to this same figure, how are the "IGHG" and "IGHA" and IGLC and IGKC counts obtained? The hg38 genome reference usually contains multiple IGHG genes (1-4) and IGHA (1-2)
- there is no information in the methods how this single count value for each antibody class was derived

- Is it possible to examine the distribution of single B cells from different patients? The authors state in the methods that they pooled samples together before library prep so this might not be possible. If so, please include analysis of this and if not please include a reference to this in the results text.

- Figure 2 - Was there differential expression of GC or other B cell signatures (from Fig1) in the TLS+ vs TLS- EC analysis? This would be an interesting avenue to explore further if possible, and could provide a stronger narrative link between Figure 1 and the rest of the manuscript.

Minor comments/concerns

- Line 71 - CXCL13, not CLCX13

- Line 93 - "CD11c" should be corrected to "ITGAX (CD11c)". This gene also marks "atypical" memory B cells (such as FCRL4+ memory B cells in the tonsil). Please include a citation for the discussed link with this gene and chronic antigen stimulation and B cell exhaustion.

- Lines 97-99 "These observations are in line with a recent study in HPV-associated HNSCC12, and suggest similarities between B-cell responses in viral and neoantigen-driven immune responses." As written this seems to refer to the isotype/class switching frequencies between the two studies - please reword to make clear precisely what the similarities are between these studies

- Line 119 - "presence of TLS was correlated with the neoantigen-rich". Would "associated" be more appropriate term than "correlated", as no actual correlation analysis was performed?

- Fig1D has "MBC" label that is different from other cluster labels throughout the figure

- Supplementary table of the marker genes for each B cell cluster would be ideal. This could either be the top (50-100) markers per cluster, or the direct output from Seurat FindAllMarkers command.

- Fig1C - it would be helpful to annotate the spatial transcriptomics figure as "lymph node" to avoid any confusion that this might be a EC tissue slice

- Did the authors examine if L1CAM is expressed by EC B cells in the scRNA-seq dataset?

- Line 248-249 "Lymphoid aggregates without any L1CAM positivity were not counted." Does this suggest that different types of TLS exist? This might be beyond the scope of the current study, but if there is any information on numbers of L1CAM- vs L1CAM+ lymphoid aggregates this would be very interesting, and not diminish from the conclusions presented.

- B cells were stained for CD27 and CD19, but it is unclear whether both or one of these markers were used for sorting single-cells - this needs to be made clear. If CD27 was included to sort cells, and naïve B cells or other CD27- populations (such as some memory populations) were not profiled, this must be discussed in relation to figure 1

- Please provide more detail is needed about quality control of scRNA data. E.g. minimum genes per cell, number of cells per gene, mitochondrial counts, and doublet inference. Was any batch

correction needed? How was the data normalised – there is a reference to SCTransform.

- Please provide citations for all software/analysis methods and datasets used in the methods in the bibliography. Eg. Seurat, SCTransform, STAR, DESeq2, King et al 2021 Sci Immunology. This last one should also be cited in the main text when referred to on line 90.

RESPONSE TO THE REFEREES - NCOMMS-21-25761A

We thank all referees for their time to review our manuscript. Their questions and suggestions have helped us to improve the manuscript substantially. We provided a point-by-point reply to all questions below this letter. We hope we have satisfactorily addressed to all concerns. If some concerns remain, we are willing to revise the manuscript further.

Sincerely, on behalf of all authors,

Nanda Horeweg M.D. Ph.D.
Clinical Epidemiologist
Department of Radiation Oncology
Leiden University Medical Center
Albinusdreef 2
2333ZA Leiden
The Netherlands
Email: n.horeweg@lumc.nl

Reviewer #1 (Remarks to the Author): with expertise in endometrial cancer

General remarks by reviewer 1:

The study reported in the manuscript by Horeweg et al., aimed to delineate the role and prognostic relevance of B-cells and tertiary lymphoid structures (TLSs) in endometrial cancer (EC). At the onset of this study, it was known that most TLSs in endometrial tumors occur in POLE-mutated (POLEmut) and mismatch repair deficient (MMRd) endometrial tumors, which are two of four major molecular subtypes of EC. The study by Horeweg et al., validates these prior observations and extends upon them to demonstrate the following:

- 1) “Antibody-secreting B-cells (in in endometrial tumors) had undergone class-switching and expressed markers of activation and exhaustion, suggesting TLS formation and an ongoing B-cell response against EC.”
- 2) “...an association of TLS with CD8+ T-cell infiltration and L1CAM overexpression..... independent of any L1CAM expression by the tumor itself.”
- 3) “L1CAM-expressing lymphoid structures appeared to be mature TLS with a germinal center, based on co-immunofluorescence and IHC for hallmark immune cell subsets. Using L1CAM expression at lymphoid structures as a marker, assessment of tumor material of 378 high-risk EC patients included in the PORTEC-3 trial revealed TLS in 19% of cases.
- 4) “...favorable prognostic impact of TLS in an independent randomized trial with high quality clinical outcome data 171 (PORTEC-3).”
- 5) The “presence of TLS remains a strong favorable prognostic factor after correction for all important clinicopathological and molecular risk factors.”

The authors conclude their “data suggests a pivotal role of TLS in outcome of EC patients, and establishes L1CAM as a simple biomarker.”

While this is an interesting study, in its current form the manuscript is at times difficult to follow particularly regarding the exploration of L1CAM as a potential biomarker for the identification of TLS in EC.

authors' reply:

We thank the reviewer for the kind words and thorough assessment of our manuscript. Based on the reviewer's specific comments, we have revised the manuscript to - among others - clarify the exploration of L1CAM as potential marker for the identification of TLS.

Specific comments reviewer 1:

Comment 1: Lines 47-49 in abstract: Please include the percentages of MSI tumor and POLE-mutant tumors that were TLS positive; also, please state that these were L1CAM-positive-TLSs.

Reply 1: 51.1% (24/47) of the POLE mutant tumors and 22.8% (29/127) of the MMRd tumors were L1CAM positive. This has been added to the abstract.

Comment 2: Lines 51-52: Last sentence of the abstract is overstated; the data in this study do not establish L1CAM as a biomarker of TLS. No data were presented regarding increased sensitivity and/or specificity of TLS identification using L1CAM-positivity compared with pathologic review of H&E sections. It would be more appropriate to state that this study highlights L1CAM is a potential biomarker for TLS in ECs.

#Reply 2: We agree that an assessment of concordance between H&E and L1CAM for the detection of TLS is indispensable to make this statement. In addition, we believe that a high interobserver agreement between pathologists when using L1CAM as a marker for mature TLS is another important requirement. Buisseret et al. (Mod Pathol 2017) demonstrated that the interobserver agreement was poor between pathologists using H&E stains to detect TLS. Hence, we conducted a concordance study and added this to the manuscript. We kindly refer to the methods (under header: Concordance study of L1CAM for the detection of mature TLS), results (header: Concordance study of L1CAM as a marker for mature TLS) and supplemental Tables S5-6 for all details. Briefly, the number of mature TLS per slide detected by H&E was systematically lower than by L1CAM. Intra-observer agreement between H&E and L1CAM was moderate (intraclass coefficient 0.79, 95% CI 0.63-0.89; kappa 0.64, SE 0.11). The interobserver agreement between 2 pathologists using L1CAM to detect mature TLS was excellent (intraclass correlation coefficient 0.94, 95% CI 0.88-0.97; kappa 0.84, SE 0.8). By adding these results we feel that the last sentence of the abstract is not overstated anymore. If the reviewer does not agree, we will adjust the statement.

Comment 3: Lines 85 & 100: More information is needed about the molecular and clinicopathologic criteria used to select the six EC tumors for scRNA-seq of B-cells, and the possible biases that the selection criteria might have introduced when defining the scRNA-seq clusters. In addition, were TLS(s) present in these six cases, and was this part of the selection criteria?

Reply 3: The six EC tumors selected for scRNA-seq were retrieved from an archival series of cryopreserved tumors. For initial collection of this archival series, tumors were collected when the diagnostic biopsy established the presence of an endometrial cancer. No other selection criteria were used, nor were the cases selected based on the presence of TLS. As discussed in more detail in reply to questions raised by reviewers 2 and 3, cells from all 6 patients were represented more or less equally across the identified clusters (see also Figure S1 of the revised manuscript).

#Comment 4: Lines 116-118: The authors refer to “L1CAM-stained whole tumor slides”. Please define “L1CAM stained.” Does this refer to slides that had L1CAM-positive staining of tumor cells, or does it refer to slides stained for L1CAM regardless of the level of L1CAM expression detected?

Reply 4: We meant that we used whole tissue slides of the PORTEC 3 trial that were stained for L1CAM by IHC, regardless of the level of expression in either tumor or TLS. We clarified this in the text (under header Clinicopathological correlations of mature TLS).

#Comment 5: What percentage of L1CAM-negative tumors have TLS?

#Reply 5: $57/273 = 20.9\%$ of L1CAM negative tumors have TLS, compared to $14/103 = 13.6\%$ of the L1CAM positive tumors (chi square test p-value 0.10). Hence L1CAM expression by tumor cells is not significantly related to L1CAM expression by follicular dendritic cells in TLS. The slightly lower proportion of L1CAM-positive tumors with mature TLS is probably a reflection of the strong relation between L1CAM and the p53abn molecular class. We have added this information to supplementary Table S9.

#Comment 6: Line 103-104 states : “TLS were more common among the neoantigen-rich MMRd and POLEmut EC-subtypes (Figure 2B).” It would be important to provide the percentage of each molecular subgroup that is TLS+, along with p-values to indicate whether the differences in TLS incidence between molecular subgroups are statistically significant.

#Reply 6: This has been added to the manuscript in the results (under header Discovery of L1CAM expression in mature TLS).

#Comment 7: Lines 108-111 states: “We followed up on this observation by performing L1CAM-immunohistochemistry (IHC) on EC samples (Figure 2D). We observed strong L1CAM staining in the GC-like structures of the TLS, which co-localized with follicular dendritic cell (FDC) marker CD21 (Figure 2E). This was independent of L1CAM overexpression by the tumor.” These statements of results are very generalized. More details are needed regarding (a) the number of EC samples assessed and the percentage of samples that were positive; (b) concordance rates of L1CAM positivity and GC-like structures of the TLS; and (c) the concordance rates of L1CAM+ in the TLS and L1CAM+ in the paired tumor sample.

#Reply 7: Regarding a) At the specific section of the results the reviewer is referring to, only a few EC samples were tested and sequential tumor sections were stained using immunohistochemistry for L1CAM and TLS hall

mark immune cell subsets markers, as displayed in Figure S5. Further down the article (from line 139 onwards) we describe that we used the biobank of a large randomized controlled trial (PORTEC 3). In 19% of the included EC samples, mature TLS were detected using L1CAM IHC (line 143).

Regarding b) We conducted a concordance study to determine this. We added this to the methods (under header Concordance study of L1CAM for the detection of mature TLS), results (under header: Concordance study of L1CAM as a marker for mature TLS) and supplemental Tables S5. It appeared that TLS with GC-like structures on H&E are very often L1CAM positive (21/24 = 87.5%). However, there were also lymphoid structures that are not clearly identifiable on H&E as a TLS with a GC, which expressed L1CAM (6/27 = 22.2%). In addition, L1CAM IHC seems to detect systematically more TLS than H&E. Concordance rates were added to the manuscript (intraclass correlation coefficient: 0.79, 95% CI 0.63-0.89; kappa 0.64, standard error 0.11).

Regarding c) We added this information to supplemental Table S9. There was no significant concordance between L1CAM expression by the tumor and L1CAM positivity of the TLS (230 were concordant (61.2%) and 146 (38.8%) were discordant, p-value = 0.10).

#Comment 8: Line 114 onwards: The rationale for exploring whether expression of L1CAM could be used as a marker for the presence of mature TLS is unclear. This part of the study seems circular in that the authors initially identified TLS in TCGA tumors by pathologic review of H&E stained tumor sections and subsequently identified differential (increased) expression of L1CAM levels in TCGA tumors with TLS. In other words, because the authors initially identified TLSs by pathologic review, it is unclear why a biomarker for TLS identification is needed. Assuming they can justify why a biomarker is warranted, the rationale for focusing on L1CAM as a potential biomarker rather than focusing on another significantly differentially expressed gene(s) highlighted in red in Figure 2C also requires justification.

#Reply 8: We improved the description of the rationale for a simple biomarker of TLS in the introduction of the manuscript and our research aims. Briefly, before this study only H&E (which has poor reproducibility), series of IHCs for immune cell subsets (more demanding and difficult to quantify) and high-tech methodologies (too resource-demanding and difficult to implement in practice) were described in the literature. Considering the potential relevance of TLS for prognostication and prediction of response to immunotherapy, a simple, cheap biomarker is warranted. When we analyzed the differential gene expressions of ECs from the TCGA (described in the results under header Discovery of L1CAM expression in mature TLS), we paid special attention to genes for which a simple functional assay, such as immunohistochemistry, exists and were significantly differentially expressed between TLS-positive and TLS-negative. We revised figure 2C to highlight L1CAM and show that it is one of the top upregulated genes in TLS positive ECs.

Reviewer #2 (Remarks to the Author): with expertise in B cells/tertiary lymphoid structures

General remarks by reviewer 2:

This is an interesting and well-presented study that investigates the roles of B cells and TLS in endometrial cancer. scRNA-seq was used to interrogate the various molecular subsets of TIL-B, revealing activated/memory B cells, GC B cells, and ASCs. TLS were associated with L1CAM expression, which in turn was investigated as a prognostic marker in a large (378 case) EC biobank (PORTEC-3). TLS were enriched in the POLE and MMRd molecular subtypes. L1CAM+ TLS were strongly favorably prognostic by multivariable analysis, independent of molecular subtype. The authors propose that TLS play a pivotal role in EC outcomes, and they propose L1CAM serves as a convenient biomarker for TLS.

Strengths:

1. B cells and TLS are the subject of much current interest.
2. scRNA-seq and spatial transcriptomics data enriches the emerging landscape of B cell phenotypes in human cancer.
3. EC provides a useful setting to evaluate B cells and TLS relative to distinct molecular subtypes.
4. Large and well annotated tissue cohort.

#General reply: We thank the reviewer for the in-depth assessment of our manuscript and the kind words on the strengths of our study. We have further improved the manuscript based on the limitations and specific suggestions for improvement outlined by the reviewer below.

#Comment 1: They provide a relatively superficial analysis of the scRNA-seq data, with no major discoveries.

#Reply 1: As discussed in detail below, we have now significantly expanded our analysis of the scRNA-seq data using both analysis of canonical marker genes, as well as by projecting the data from the EC scRNA-seq data onto a reference map derived from the tonsillar scRNA-seq data from King et al. These analyses support

the observation that EC-infiltrating B cells are characterized by ongoing GC reactions, including cycling B cells and differentiation towards plasmablasts.

#Comment 2: The POLE and MMRd subtypes are known to be immunologically hot (including B cells and plasma cells; PMID: 30523022), therefore it is not surprising they are enriched for TLS.

#Reply 2: Using a molecularly characterized and well-annotated cohort, we - for the first time - demonstrate that the presence of TLS in EC is not merely a representation of tumor 'hotness', as i) not all ultra-mutated POLE tumors form TLS (Figure 2B and 3), nor do ii) TLS occur solely in 'hot' tumors with high infiltration of T and B cells (Figure 3). Indeed, more complex genomic alterations such as co-occurring *TP53* and *POLE/MMRd* mutations were significantly associated with TLS (Table S8). Our data thus provide novel insight into the complexity of TLS formation in human tumors.

#Comment 3: TLS have been shown to associate with favorable prognosis in many cancers, therefore this finding is not novel, apart from providing a convincing example in EC.

#Reply 3: TLS have indeed been shown to be associated with a favorable prognosis in a number of cancers, but the maturity of the TLS seems to be relevant for the prognostic impact of TLS. Gunderson et al. (Oncoimmunol 2021) showed that pancreatic cancer patients with mature TLS (with a germinal center) had enhanced humoral immunity and diminished TGF-beta signaling leading to a significantly longer survival compared to patients having TLS without a germinal center. Our study is the first to identify L1CAM as a simple biomarker for the identification of mature TLS. We conducted an additional concordance study of TLS quantification by L1CAM and H&E and found that L1CAM was never expressed in TLS that did not morphologically look like mature TLS. This implies that potentially, L1CAM is a biomarker for clinically relevant TLS. This has been added to the Discussion (end of first paragraph).

#Comment 4: The authors provide only a superficial validation of L1CAM as a surrogate biomarker for TLS. There is no comparison between the number of TLS one captures with canonical markers vs. L1CAM. And are they the same TLS with the same features? They show only one representative image. Moreover, their analysis is restricted to EC, leaving it an open question whether L1CAM is relevant to other cancers.

#Reply 4: During revision of the manuscript, work by Heesters et al. [PMID: 34424268] identified L1CAM as a marker for follicular dendritic cells, but not marginal reticular cells (MRCs), or fibroblastic reticular cells (FRCs). These data support our observations on L1CAM expression by FDCs, and indicate L1CAM could be a useful marker for mature (FDC+) TLS across cancer types. We have included these observations and appropriate corresponding references in the revised manuscript.

In addition, we have performed a concordance study to better validate L1CAM as a surrogate marker for mature TLS. In short, this sub-study showed that the number of mature TLS per slide detected using L1CAM was systematically higher than by H&E and the interobserver agreement between 2 pathologists using L1CAM to detect mature TLS was excellent (intraclass correlation coefficient 0.94, 95% CI 0.88-0.97; kappa 0.84, SE 0.8).

Suggestions for improvement:

#Suggestion 1: For the scRNA-seq expts, which molecular subtypes of EC tumors were used? The authors claim that cluster 1, 2 and 3 presented in Figure 1A correspond to activated/memory B cells, cycling/GC B cells, and antibody-secreting cells respectively. With the differentially expressed genes highlighted in Fig. 1B, I am convinced that cells in cluster 3 are indeed antibody-secreting cells. However, the identity of the other two clusters remains unclear. The authors should provide a more in-depth analysis of canonical marker genes expected in each of these cell types. Perhaps using the DotPlot() function from Seurat to plot expression of canonical marker genes across clusters would be more convincing. The study by King et al, cited in the Methods section, does provide an extensive list of marker genes that could be used here [PMID: 33579751].

#Reply to suggestion 1: As suggested by the reviewer, we have performed a more in-depth analysis of the scRNA-seq data (Figure 1). Following clustering (Figure 1A), we used the DotPlot() function of Seurat as suggested by the reviewer to analyze a series of canonical markers derived from suggestions of reviewer 3 and/or from the study by King et al. (Figure 1B). Clusters of B cells were characterized by: naïve B cell genes including *SELL* and *TCL1A* (cluster 1); (pre-)Germinal Centre (GC)-like and Cycling B cell genes *BHLHE40*, *BCL6*, *MKI67* and *HMGB2* (cluster 2); and plasma cell genes *PRDM1*, *XBPI*, *MZB1*, and *SSR4* (Cluster 3). Individual patients were similarly represented within each cluster (Figure S1). To confirm the identity of cells in each cluster, we next mapped the EC B cell dataset onto the reference dataset of tonsillar B cells from King et al. (Figure 1C) using the FindTransferAnchors() and MapQuery() functions from Seurat. This projection revealed Cluster 1 cells resembled naïve B cells, cluster 2 a mixture of naïve, activated, cycling, memory B cells and plasmablasts, and cluster 3 naïve B cells and plasmablasts (UMAP projection in Figure 1C and quantified in 1D). These analyses have been incorporated into the revised manuscript.

#Suggestion 2: Besides the marker gene analysis presented in Fig. 1B, the authors used spatial transcriptomics to help assign a cell type to each of the three clusters identified by scRNA-seq. However, the explanations provided in the manuscript (including the ones in the Methods section) are too vague, making it hard to understand what was actually done.

#Reply to suggestion 2: Based on the suggestion of reviewer 3, we have opted to remove the spatial transcriptomics analysis in favor of projecting the data onto the single cell dataset by King et al. as described above and depicted in Figure 1 of the revised manuscript.

#Suggestion 3: In Figure 1D, the authors argue that this cluster represents activated/memory B cells that might be exhausted given that they express both HLA-DRA and CD11C. However, very few cells seem to actually express these two genes, and the cluster seem rather heterogeneous based on the expression of the characteristic marker genes shown in Figure 1B. It might be useful to characterize that cluster further by re-analyzing it similarly to what was done for the antibody-secreting cells in the next panel.

#Reply to suggestion 3: As discussed above, we have now reanalyzed the data to clarify the identity of the cells within clusters 1-3.

#Suggestion 4: If you redo the analysis presented in Figure 1E but masking the immunoglobulin genes, are there any other genes that emerge as differentially expressed between the clusters? Could they help identify plasma cell subtypes in EC? For instance, do the IgA+ plasma cells appear to be immunosuppressive as has been reported in murine studies [PMID 29144460][PMID 25924065]?

#Reply to suggestion 4: We have included the differential expression analysis of the plasma cell clusters as Table S2 of the revised manuscript. The number of genes differentially expressed between these clusters was minimal and we did not observe hallmark immunosuppressive genes such as PDL1, IL-10 or TGF within the IgA+ cells. It should however be noted that human IgA+ plasma cells are not always immunosuppressive and have also been shown to promote immunity and T cell responses in for instance ovarian cancer (PMID: 33536615).

#Suggestion 5: Ideally, all codes as well as processed scRNA-seq data should be made available for download using a GitHub repository or Zenodo. If possible, the raw data should also be made public in an appropriate repository or at least be made available upon request. Similarly, the authors should add Supplementary Tables with the differentially expressed genes identified in each comparison.

#Reply to suggestion 5: The raw data was deposited prior to submission (GSE180091). No custom code was used for analysis. This has been added to the Data Availability and Code availability sections of the manuscript. Supplementary Tables with the differentially expressed genes identified in each comparison have been added as described above.

#Suggestion 6: In Fig. 2A, exactly what criteria were used to quantify TLS in H&E stained sections? In Figure 2B, a Fisher's exact test should be used to assess statistical significance.

#Reply to suggestion 6: A description of the criteria for TLS quantification of mature TLS using H&E has been added to the methods (under the header Quantification of LICAM-positive mature TLS using H&E and LICAM IHC).

We tested the significance of the differences in the distribution of TLS presence across the molecular classes using the Fisher-Freeman-Halton exact test (the equivalent of the Fisher's exact test for crosstables larger than 2x2) and added this to the methods (Statistical analysis) and the results (under the header Discovery of LICAM expression in mature TLS).

#Suggestion 7: The observation made in Figure 2C is interesting. However, no comments are made on the genes upregulated in TLS-negative tumors. Are they linked to tumour progression? Moreover, rather than cherry picking the genes to highlight, it would be preferable to formalize the analysis using GSEA or the like. In terms of graphical representation, encoding the p-value with the size of the dot is difficult to read given the extent of overplotting. Consider using arrows to signal what is up and down-regulated, and use colors to highlight the most significant genes.

#Reply to suggestion 7: We performed a GSEA analysis of genes differentially expressed between TLS-positive and TLS-negative TCGA UCEC cases (Table S3) using GO terms for biological process (Table S4 and Figure S4). As expected, TLS-positive cases were enriched for genesets associated with lymphocyte activation and adaptive immune responses, whereas genesets associated with TLS-negative cases included e.g. (microtubule-based) cell motility genesets. We have also reduced the overplotting in Figure 2C to focus on the identification of LICAM, with additional details on the differentially expressed genes and genesets moved to the supplementary data.

#Suggestion 8: With regard to the data presented in Figure 2D and E, the authors should comment on the prevalence of these L1CAM-TLS since they mention in the Methods that ‘Lymphoid aggregates without any L1CAM positivity were not counted’. Similarly, according to Supplementary Figure 2, L1CAM+ TLS show all the features of a typical TLS. Could this be quantified? That is, out of the L1CAM+ TLS stained with CD4, CD8, Bcl6, CD20 panel, how many have a typical architecture? Such quantifications are required before calling L1CAM a TLS biomarker.

#Reply to suggestion 8: The prevalence of TLS that express L1CAM is estimated at 19% based on the 378 high-risk endometrial cancer patients participating in the randomized PORTEC 3 trial (Results under header Clinicopathological correlations of mature TLS). Our expert gynecologist (T.B.) who quantified the TLS of all these cases reports that all L1CAM-positive TLS also had the morphology of a mature TLS. All the cases with L1CAM expressing TLS that we stained for Bcl6, CD20, CD4 and CD8 on subsequent tissue slides (supplemental figure S5 is a representative example) had the hallmark features of mature TLS. However, these were too few cases to deduct a formal quantification. Moreover, there is no consensus in the literature on the quantification of the immunostains in this context. We prefer not to perform a post-hoc analysis with a self-defined quantification algorithm. Nonetheless, the point of the reviewer that additional quantifications to enforce the potential of L1CAM as a biomarker for mature TLS is well taken. We therefore conducted a concordance study wherein we quantified mature TLS using L1CAM and H&E. We kindly refer to the methods (Concordance study of L1CAM for the detection of mature TLS), results (Concordance study of L1CAM as a marker for mature TLS) and supplemental Tables S5-6 for all details. Briefly, the number of mature TLS per slide detected by H&E was systematically lower than by L1CAM. Intra-observer agreement between H&E and L1CAM was moderate (intraclass coefficient 0.79, 95%CI 0.63-0.89; kappa 0.64, SE 0.11). The interobserver agreement between 2 pathologists using L1CAM to detect mature TLS was excellent (intraclass correlation coefficient 0.94, 95%CI 0.88-0.97; kappa 0.84, SE 0.8).

#Suggestion 9: In Fig. S3, exactly how were TLS assessed (i.e. what were the precise scoring criteria)? In Figure 3D and E, no p-value is reported.

#Reply to suggestion 9: We clarified the wording to the TLS quantification by L1CAM and added a description of the quantification of TLS on H&E in the methods (Quantification of L1CAM-positive mature TLS using H&E and L1CAM IHC). We added p-values to figure 3D and E.

#Suggestion 10: Figure 4B/C are barely commented by the authors. What is an acceptable c-index? Can interesting conclusions be drawn by looking at the relative importance of the variable? Moreover, in Figure 4C, the authors should replace these pie charts by bar plots to make the comparison easier for the reader [PMID 24645191].

#Reply to suggestion 10:

We added a better description of Figure 4B/C to the results (Independent prognostic value of mature TLS). There is, to our knowledge no universal consensus on an acceptable c-index, but in general a c-index under 0.70 is considered insufficient and a c-index above 0.80-0.85 is rarely achieved. The purpose and envisioned implementation of the model generally dictates the requirements for model fit and other parameters. Here, we report the c-index alongside with other indicators of model fit to show that prediction of time to recurrence improves with the addition of TLS. The c-index should be interpreted as relative measure to compare models, rather than an absolute value as we do not regard our model as a prediction tool.

Yes, we do think that interesting hypothesis-generating conclusions can be drawn from the relative importance of the predictors in the 3 models, and we added this to the results (Independent prognostic value of mature TLS) and discussion (end of first paragraph). Briefly, TLS seem to convey a part of the prognostic impact that has previously been attributed to the EC molecular class. We hypothesize that the prognostically favorable impact of mis-match repair deficiency originates at least partly in an active immune response against EC. MMRd ECs without TLS have relatively poor prognosis (also presented in figure 3E).

According to the article cited by the reviewer layered bar charts would be the best choice comparing the contribution of each predictor to time to recurrence. We've plotted the bar charts (sample provided below). Although the contribution to each of the 3 models for e.g. LVSI now easier to deduct, the relation between the different predictors within each of the 3 models is lost with this visualization. Although we acknowledge the value of the referred article, we think that it does not apply in this situation because our data consists of relative amounts instead of absolute quantities. We will keep the pie charts and expect that they will be appreciated by many as this is the case for similar pie charts in our previous publication on CD8 TILs in EC (Horeweg et al. 2020 CIR).

#Suggestion 11: The authors should discuss why their findings differ from those of Talhouk [PMID: 30523022] with respect to the relative prognostic effects of TIL versus molecular subtype (including TIL-B).

#Reply to suggestion 11: The results of the study by Talhouk et al. do not really differ from ours, it is rather the interpretation of the results that is different. We both identified that TIL-Ts in EC are more common in POLE and MMRd and that B-cell densities within tumor is quite low (Figure 3A) and in the stroma a bit more abundant, which is more prominent in POLE and MMRd EC. In our previous study on T cells (Horeweg et al, CIR 2020) we showed that patients with dense T cell infiltrates have a better prognosis, and Talhouk et al. found this as well; citation “Within MMRd cases, CD8+CD3+ and CD3+CD8+ TILs were positively associated with DSS (P = 0.04 and 0.02, respectively). Where our results seem different, is whether or not the favorable prognostic impact is independent or not; Talhouk et al. do not find this, while we do. Talhouk et al. interpret the absence of proof as proof of absence. However, there are several factors that may have contributed to the absence of proof: i) their data comes from clinical cohorts is biased by confounding by indication which weakens the relation between prognostic predictors and oncological outcomes, ii) insufficient numbers of events to support the multivariable analysis (eg 13 predictors for 68 DSS events) iii) the chosen endpoints OS, DSS and PFS are all not ‘pure’ measures of recurrence, which introduces noise in the relation with the predictor, iv) use of a dichotomized dummy for immune cell densities and an immune cluster may not be the best way to test the relationship with oncological outcome, etc ect. We have addressed to this in our previous publication (Horeweg et al. CIR 2020), so we will not repeat that in the discussion of this manuscript. In addition, the focus of this manuscript is on TLS, which are not studied by Talhouk et al.

Reviewer #3 (Remarks to the Author): with expertise in B cell immunology, scRNAseq

#General remarks

This study from Horeweg et al examines the composition, phenotypic markers and link with disease progression of tertiary lymphoid structures (TLS) in endometrial cancer (EC). The authors first examine the B cell compartment of ECs using single-cell RNA-seq (scRNA) and report distinct B cell phenotypes associated with activation, GC maturation or antibody production. They then leverage biobanked tissue resources to quantify the frequency of TLS structures in different classes of EC, and report a particular association between TLS and the POLEmut and MMRd ECs. Differential gene expression of TLS positive and TLS-negative ECs identified L1CAM as a histological marker of TLS, which they then used to build a prognostic model. From this, the authors propose that the existence of TLS is associated with favourable patient outcomes.

This is a high quality study that I found to be very well written and reasoned, and it tackles an important and interesting area of cancer immunology. As well as reporting a convincing association between TLS and patient outcome, the transcriptomic profiling of tumour-associated/TLS B cells is a valuable resource for the scientific community to help understand the formation, maintenance and function of TLS, which remain poorly understood. Furthermore, while TLS are known to play an important role in mediating immune responses to cancer (as convincingly presented and discussed in this manuscript), they are also a key feature in the pathology of varied autoimmune diseases (eg lupus, rheumatoid arthritis, Sjogrens syndrome). Thus, the dissection of the structure and function of TLS is relevant beyond EC and will be of interest to many immunologists.

My only major reservation relates to the precise phenotypic annotation of B cell states by scRNA-seq, which can be addressed with additional analysis and discussion.

#Reply to general remarks: We thank the reviewer for the kind words and the suggestions for additional analyses and discussion. As discussed in detail below, we have revised the manuscript accordingly to provide the requested insight into the B cell states examined by scRNA-seq.

Major comment 1: Some of the markers used to describe or annotate B cell scRNA-seq clusters C1 and C2 should be re-examined. (The markers used to annotate C3 as antibody secreting cells/plasma cells are very convincing). The spatial mapping generally supports the other annotations, but some of the current markers used are not particularly convincing. I suggest that the authors examine the following:

- o Cell cycle stage (G1, G2M, S) – HMGB2 (a marker for C2 cluster) is most highly expressed by proliferating/cycling B cells. These may be cycling B cells (most likely GC B cells) that cluster separately very distinctly from other GC B cells that could be contained within C1

- o Line 87 - HMGB2, BHLHE40 and VIM are not classical markers for GC B cells. Please examine BCL6 as a canonical GC B cell marker, and other transcriptional markers like GMDS, BACH2, LMO2, SERPINA9, AICDA, ISG20.
- o Some markers for C1 such as EZR and FOXP1 are also expressed by GC B cells. It is often helpful to visualise key marker genes on the single cell UMAP plot (FeaturePlot in Seurat) and I recommend plotting some of the key canonical marker genes for different B cell states in this fashion.
- o Are there any memory B cells present as part of C1? Please examine TNFRSF13B (most expressed by memory cells) and TCL1A (expressed by naïve and GC B cells, but not memory or ASCs)
- o Other markers I would suggest to examine include for CD69 and JUN for activated B cells, FCER2 for basal naïve B cells.

#Reply to major comment 1: As suggested by the reviewer, we have performed a more in-depth analysis of the scRNA-seq data (Figure 1). Following clustering (Figure 1A), we used the DotPlot() function of Seurat to analyze *HMGB2*, *BCL6*, *GMDS*, *BACH2*, *LMO2*, *SERPINA9*, *AICDA*, *ISG20*, *TNFRSF13B*, *TCL1A*, *CD69*, *JUN* and *FCER2*, as well as a number of markers from the study by King et al. (Figure 1B). Clusters of B cells were characterized by: naïve B cell genes including *FCER2* and *TCL1A* (cluster 1); (pre-)Germinal Centre (GC)-like and Cycling B cell genes *BHLHE40*, *BCL6*, *MKI67* and *HMGB2* (cluster 2); and plasma cell genes *PRDM1*, *XBPI1*, *MZB1*, and *SSR4* (Cluster 3). Individual patients were similarly represented within each cluster (Figure S1).

#Major comment 2: Please provide a more complete list of marker genes per cluster as a supplementary table. Either the top 50 or 100 genes per cluster, or the direct output from Seurat's FindAllMarkers command.

#Reply to major comment 2: All output of the Seurat's FindAllMarkers command for all comparisons throughout the manuscript have now been included as supplemental tables (Table S1 and S2).

#Major comment 3: I would recommend examining/visualising the antibody class expression (IgM, IgD, IgG, IgA, IgE) of all B cell clusters, not just ASCs as presented in Fig1E. This can be very useful to help annotate different B cell phenotypes and may reveal insights into whether memory B cells of different classes may be interesting or relevant to TLS/EC.

#Reply to major comment 3: We have included visualization of antibody class genes *IGHG1-4*, *IGHA1-2*, *IGHE*, *IGHD*, *IGHM*, *IGLC2*, *IGLC3* and *IGKC* for all clusters (Figure S2), and within the subsetted and reclustered plasmablast cells (Figure S3). Other IGLC genes, such as *IGLC1*, were not detected within our dataset. Summarized, we found distinct clusters consistent with B-cell class switching, and largely based on differential expression of heavy chain *IGHG*, *IGHA* genes and the mutually exclusive light chain *IGLC* and *IGKC* genes (Fig. 1E). Differential gene expression analysis revealed these IgG and IgA plasmablasts were transcriptomically similar (Table S2).

#Major comment 4: Was the tonsil scRNA-seq data from King et al 2021 (that was used as a reference for spatial transcriptomics) also integrated with the TLS EC B cell scRNA dataset? If the B cell annotations from King et al were transferred/projected on to the data from this study (or vice versa) does this support the annotations presented in Figure 1? In my opinion, this would be more convincing evidence to support the annotations that the comparison with the spatial mapping that was performed (although this is also very valuable).

#Reply to major comment 4: We mapped the EC B cell dataset onto the reference dataset of tonsillar B cells from King et al. (Figure 1C) using the FindTransferAnchors() and MapQuery() functions from Seurat. This projection revealed Cluster 1 cells resembled naïve B cells, cluster 2 a mixture of naïve, activated, cycling, memory B cells and plasmablasts, and cluster 3 naïve B cells and plasmablasts (UMAP projection in Figure 1C and quantified in 1D). These analyses are consistent with the analysis of the canonical markers described above and have been incorporated into the revised manuscript. In line with the reviewer's suggestion, we have chosen to use this integration to support the annotations derived from canonical marker analysis instead of the spatial transcriptomics analysis.

#Major comment 5: Line 161-163 “The antibody secreting B-cells had undergone class-switching and expressed markers of activation and exhaustion, suggesting TLS formation and an ongoing B-cell response against EC.” As presented, the ASCs (plasma cells) do not consistently express the activation markers or exhaustion (CD11c) this should be re-written. Do the authors mean the activated/memory B cells?

#Reply to major comment 5: The reviewer is correct and we have rewritten this accordingly.

#Major comment 6: The authors may like to refer to a recent analysis of the same human lymph node spatial transcriptomic dataset that they examined (Kleshchevnikov et al 2020 bioRxiv 10.1101/2020.11.15.378125). I believe that their conclusions broadly support the analysis presented here.

#Reply to major comment 6: As we have chosen to use the scRNA-seq dataset integration to support the cell annotation, we have chosen at this time not to include this specific reference. We agree with the reviewer that the conclusions presented by Kleshchevnikov et al. are in line with our observations.

#Major comment 7: Fig 1E - Did the authors examine differential expression between these clusters? I am not convinced that this clustering analysis represents any major biological insight. The same information (ie that ASCs contain both IgG and IgA populations) could be presented by visualising these genes on the same UMAP as in Fig1A (with the advantage of presenting information about the isotype of other B cell populations as well)

#Reply to major comment 7: All differential expression analyses have been included as supplementary tables using the output of Seurat's FindAllMarkers() command.

#Major comment 8: Relating to this same figure, how are the “IGHG” and “IGHA” and IGLC and IGKC counts obtained? The hg38 genome reference usually contains multiple IGHG genes (1-4) and IGHA (1-2) – there is no information in the methods how this single count value for each antibody class was derived

#Reply to major comment 8: In line with the reviewer's suggestion on the analysis of antibody class expression (IgM, IgD, IgG, IgA, IgE) for all B cells, we have now depicted these genes individually.

#Major comment 9: Is it possible to examine the distribution of single B cells from different patients? The authors state in the methods that they pooled samples together before library prep so this might not be possible. If so, please include analysis of this and if not please include a reference to this in the results text.

#Reply to major comment 9: We have included an analysis of the single B cells from different patients as Figure S1. In brief, all clusters were represented per patient, although the frequencies differed between patients.

#Major comment 10: Figure 2 - Was there differential expression of GC or other B cell signatures (from Fig1) in the TLS+ vs TLS- EC analysis? This would be an interesting avenue to explore further if possible, and could provide a stronger narrative link between Figure 1 and the rest of the manuscript.

#Reply to major comment 10: We performed a GSEA analysis of genes differentially expressed between TLS-positive and TLS-negative TCGA UCEC cases (Table S3) using GO terms for biological process (Table S4 and Figure S4). TLS-positive cases were enriched for genesets associated with lymphocyte activation and adaptive immune responses, including a.o. GO:0019724 (B cell mediated immunity) and GO:0042113 (B cell activation).

Minor comment 1: Line 71 – CXCL13, not CLCX13

Reply to minor comment 1: This has been corrected

#Minor comment 2: Line 93 – “CD11c” should be corrected to “ITGAX (CD11c)”. This gene also marks “atypical” memory B cells (such as FCRL4+ memory B cells in the tonsil). Please include a citation for the discussed link with this gene and chronic antigen stimulation and B cell exhaustion.

Reply to minor comment 2: The reference to CD11c has been removed based on the in-depth reanalysis of the data.

#Minor comment 3:- Lines 97-99 “These observations are in line with a recent study in HPV-associated HNSCC12, and suggest similarities between B-cell responses in viral and neoantigen-driven immune responses.” As written this seems to refer to the isotype/class switching frequencies between the two studies – please reword to make clear precisely what the similarities are between these studies

Reply to minor comment 3: This has been reworded.

#Minor comment 4: Line 119 – “presence of TLS was correlated with the neoantigen-rich”. Would “associated” be more appropriate term than “correlated”, as no actual correlation analysis was performed?

#Reply to minor comment 4: Agree, this has been changed.

#Minor comment 5: Fig1D has “MBC” label that is different from other cluster labels throughout the figure

#Reply to minor comment 5: The labels have been adjusted in the revised figures.

#Minor comment 6: Supplementary table of the marker genes for each B cell cluster would be ideal. This could either be the top (50-100) markers per cluster, or the direct output from Seurat FindAllMarkers command.

#Reply to minor comment 6: All output of the Seurat's FindAllMarkers command for all comparisons throughout the manuscript have now been included as supplemental tables (Table S1 and S2).

#Minor comment 7: Fig1C –it would be helpful to annotate the spatial transcriptomics figure as “lymph node” to avoid any confusion that this might be a EC tissue slice

#Reply to minor comment 7: As we have chosen to use the scRNA-seq dataset integration to support the cell annotation, this data has been removed.

#Minor comment 8: Did the authors examine if L1CAM is expressed by EC B cells in the scRNA-seq dataset?

#Reply to minor comment 8: We examined L1CAM expression in our dataset, but EC B cells did not express this gene (at detectable levels). L1CAM was also not expressed by B cells in the dataset from King et al. Accordingly, during revision of the manuscript, work by Heesters et al. [PMID: 34424268] also identified L1CAM as a marker for follicular dendritic cells, but not marginal reticular cells (MRCs), or fibroblastic reticular cells (FRCs). These data support our observations on L1CAM expression by FDCs, and indicate L1CAM as a marker for mature (FDC+) TLS.

#Minor comment 9: Line 248-249 “Lymphoid aggregates without any L1CAM positivity were not counted.”

Does this suggest that different types of TLS exist? This might be beyond the scope of the current study, but if there is any information on numbers of L1CAM- vs L1CAM+ lymphoid aggregates this would be very interesting, and not diminish from the conclusions presented.

#Reply to minor comment 9: We conducted a concordance study wherein intra- and interobserver variation of TLS quantification using H&E and L1CAM stained slides was assessed. We refer to the methods (Concordance study of L1CAM for the detection of mature TLS), results (Concordance study of L1CAM as a marker for mature TLS) and supplemental Tables S5-6 for all details. Uncertainty in the distinction between a lymphoid aggregate and a TLS was reported in 26% using H&E-stained slides, while L1CAM expression was never found in lymphoid aggregates that did not have the morphology of a mature TLS. This suggests that L1CAM expression in TLS is probably very specific to mature TLS with a GC. Lymphoid aggregates other than mature TLS (not rounded and/or organized) do exist but probably don't express L1CAM.

#Minor comment 10: B cells were stained for CD27 and CD19, but it is unclear whether both or one of these markers were used for sorting single-cells – this needs to be made clear. If CD27 was included to sort cells, and naïve B cells or other CD27- populations (such as some memory populations) were not profile, this must be discussed in relation to figure 1

#Reply to minor comment 10: Only CD19 was used to sort the single cells. This has been clarified in the materials and methods section.

#Minor comment 11: Please provide more detail is needed about quality control of scRNA data. E.g. minimum genes per cell, number of cells per gene, mitochondrial counts, and doublet inference. Was any batch correction needed? How was the data normalised – there is a reference to SCTransform.

#Reply to minor comment 11: The materials and methods section has been updated to provide more detail on these analyses.

#Minor comment 12: Please provide citations for all software/analysis methods and datasets used in the methods in the bibliography. Eg. Seurat, SCTransform, STAR, DESeq2, King et al 2021 Sci Immunology. This last one should also be cited in the main text when referred to on line 90.

#Reply to minor comment 12: We have added citations for all software/analysis methods and datasets.

Reviewers' Comments:

Reviewer #1:

Remarks to the Author:

The authors have provided satisfactory responses to all my previous comments. I would strongly recommend that the following text (from the authors' response to comment-3) be added to the "Patient Material" subsection of the methods:

"For initial collection of this archival series, tumors were collected when the diagnostic biopsy established the presence of an endometrial cancer. No other selection criteria were used, nor were the cases selected based on the presence of TLS."

Reviewer #2:

Remarks to the Author:

The manuscript is significantly improved in response to the reviews. The narrative is more straightforward, and the key messages are clear. The added description of the pathology validation study for L1CAM as a TLS biomarker is helpful. My only remaining concern is whether the findings are significantly noteworthy for a broad audience outside the endometrial cancer (EC) community. The favorable prognostic effect of TLS -- while very clear in this study -- is not novel, even in EC now (their ref. 8). The scRNAseq study did not yield significant new insights into TIL-B phenotypes, at least not that are discussed. L1CAM is an interesting and promising TLS biomarker, but is it an improvement on CD21 or CD23, which are already widely used to define TLS? Moreover, now that L1CAM has been shown to be expressed by FDCs, the novelty of its association with TLS is diminished. Furthermore, the authors have not validated L1CAM outside of EC. In summary, this is now a very solid study with importance to the gynecology community but arguably not the larger cancer immunology community.

Reviewer #3:

Remarks to the Author:

I am satisfied that my comments have been adequately addressed. I commend the authors on the much improved single-cell RNA-seq analysis figures.

RESPONSE TO THE REFEREES - NCOMMS-21-25761B

Reviewer #1 (Remarks to the Author): with expertise in endometrial cancer

The authors have provided satisfactory responses to all my previous comments. I would strongly recommend that the following text (from the authors' response to comment-3) be added to the "Patient Material" subsection of the methods:

"For initial collection of this archival series, tumors were collected when the diagnostic biopsy established the presence of an endometrial cancer. No other selection criteria were used, nor were the cases selected based on the presence of TLS."

authors' reply: We thank the reviewer and are happy that all concerns have adequately been addressed. We have added the phrase to the manuscript.

Reviewer #2 (Remarks to the Author): with expertise in B cells/tertiary lymphoid structures

The manuscript is significantly improved in response to the reviews. The narrative is more straightforward, and the key messages are clear. The added description of the pathology validation study for L1CAM as a TLS biomarker is helpful. My only remaining concern is whether the findings are significantly noteworthy for a broad audience outside the endometrial cancer (EC) community. The favorable prognostic effect of TLS -- while very clear in this study -- is not novel, even in EC now (their ref. 8). The scRNAseq study did not yield significant new insights into TIL-B phenotypes, at least not that are discussed. L1CAM is an interesting and promising TLS biomarker, but is it an improvement on CD21 or CD23, which are already widely used to define TLS? Moreover, now that L1CAM has been shown to be expressed by FDCs, the novelty of its association with TLS is diminished. Furthermore, the authors have not validated L1CAM outside of EC. In summary, this is now a very solid study with importance to the gynecology community but arguably not the larger cancer immunology community.

authors' reply: We thank the reviewer for the re-review and are happy that the manuscript is deemed much improved and our study is considered very solid and important.

As the reviewer points out rightly, we have only focused on endometrial cancer in this study. We believe though that our results are relevant outside the endometrial cancer community. We are supported in this belief by Reviewer 3, who wrote: "*the transcriptomic profiling of tumour-associated/TLS B cells is a valuable resource for the scientific community to help understand the formation, maintenance and function of TLS, which remain poorly understood. Furthermore, while TLS are known to play an important role in mediating immune responses to cancer (as convincingly presented and discussed in this manuscript), they are also a key feature in the pathology of varied autoimmune diseases (eg lupus, rheumatoid arthritis, Sjogrens syndrome). Thus, the dissection of the structure and function of TLS is relevant beyond EC and will be of interest to many immunologists.*" Also, from a more clinical point of view, we are convinced that TLS are a high-potential prognostic and predictive biomarker in many forms of cancer. For example in melanoma (Cabrita et al. Nature 2020), sarcoma (Petitprez et al. Nature 2020) and renal cell carcinoma (Helmink et al. Nature 2020). Our discovery of L1CAM as a simple marker of mature TLS, may well become very relevant for the allocation of immunotherapy to different groups of oncological patients.

Reviewer #3 (Remarks to the Author): with expertise in B cell immunology, scRNAseq

I am satisfied that my comments have been adequately addressed. I commend the authors on the much improved single-cell RNA-seq analysis figures.

authors' reply: We thank the reviewer and are happy that all concerns have adequately been addressed.